# Colloidal zinc oxide-copper(I) oxide nanocatalysts for selective aqueous photocatalytic carbon dioxide conversion into methane

Kyung-Lyul Bae [1], Jinmo Kim [1], Chan Kyu Lim [1], Ki Min Nam[2] & Hyunjoon Song [1]

Developing catalytic systems with high efficiency and selectivity is a fundamental issue for photochemical carbon dioxide conversion. In particular, rigorous control of the structure and morphology of photocatalysts is decisive for catalytic performance. Here, we report the synthesis of zinc oxide-copper(I) oxide hybrid nanoparticles as colloidal forms bearing copper (I) oxide nanocubes bound to zinc oxide spherical cores. The zinc oxide-copper(I) oxide nanoparticles behave as photocatalysts for the direct conversion of carbon dioxide to methane in an aqueous medium, under ambient pressure and temperature. The catalysts produce methane with an activity of 1080 $\mu mol\, g_{cat}^{-1}\, h^{-1}$, a quantum yield of 1.5% and a selectivity for methane of >99%. The catalytic ability of the zinc oxide-copper(I) oxide hybrid catalyst is attributed to excellent band alignment of the zinc-oxide and copper(I) oxide domains, few surface defects which reduce defect-induced charge recombination and enhance electron transfer to the reagents, and a high-surface area colloidal morphology.

[1] Department of Chemistry, Korea Advanced Institute of Science and Technology, and Center for Nanomaterials and Chemical Reactions, Institute for Basic Science (ibs), Daejeon 34141, Republic of Korea. [2] Department of Chemistry, Mokpo National University, Jeonnam 58554, Republic of Korea. Kyung-Lyul Bae and Jinmo Kim contributed equally to this work. Correspondence and requests for materials should be addressed to K.M.N. (email: namkimin.chem@gmail.com) or to H.S. (email: hsong@kaist.ac.kr)

There has been intensive research on direct carbon dioxide (CO$_2$) conversion reactions via photochemical, electro-chemical, and biological approaches[1–3]. A photochemical method using sun light in aqueous solutions is regarded as a leading potential approach due to the prospect of using free and plentiful solar energy without damaging the environment[4–6]. Titanium dioxide (TiO$_2$) is a representative photocatalytic material for this purpose, due to its effective charge separation ability, abundance, and low environmental toxicity[7, 8]. The addition of co-catalysts such as platinum (Pt) and copper (Cu) can enhance the catalytic activity[9–11]. However, these TiO$_2$-based hybrid catalysts mostly generate hydrogen (H$_2$) rather than car-bon species from carbonated water[12], because the electrochemical reduction potentials of water to H$_2$ (−0.41 V vs. normal hydrogen electrode (NHE)) and CO$_2$ to reduced species (−0.58 to −0.24 V vs. NHE) are in a similar range[9]. Consequently, a novel strategy for increasing selectivity would be helpful for enhancing CO$_2$ conversion reactions.

Copper oxides are p-type semiconductors with narrow bandgaps (CuO, $E_g$ = 1.35 to 1.7 eV; Cu$_2$O, $E_g$ = 1.9 to 2.2 eV) and have been employed in pigments, solar cells, electrodes, and catalysts for organic reactions[13, 14]. In particular, their favorable light absorption in the visible range enables copper oxides to be photocatalytic materials. The formation of hybrids with TiO$_2$ can form p–n type junctions, which exhibit better charge separation and enhanced activity for photocatalytic CO$_2$ reduction[13]. Schaak et al.[15] deposited TiO$_2$ onto Cu$_3$N nanocubes at high temperature to yield hollow TiO$_{2−x}$N$_x$-CuO nanocubes, which showed high conversion of CO$_2$ to CH$_4$[15]. Ye et al.[16] synthesized porous TiO$_2$-Cu$_2$O nanojunction materials, which exhibited a large enhancement in CH$_4$ evolution activity. Although the proper combination of semiconductor and co-catalyst is essential, the structure and morphology (e.g. size, shape, and surface structure of each domain and their interfaces) is also critical in determining the catalytic properties. Rigorous control of these factors is critical for designing a photocatalyst with optimal performance[17].

Here, we select the combination of Zn(II) oxide and Cu(I) oxide for effective photocatalytic CO$_2$ conversion. Zn-Cu oxides are known from their use as photocatalysts for dye degradation[18, 19]. We expect that Zn and Cu oxides will also be an excellent photo-catalyst for CO$_2$ reduction, due to the fact that CO$_2$ species are readily adsorbed on the surface sites of metal oxides[20, 21]. Guided by this inspiration, we are able to successfully grow Cu$_2$O single-crystalline nanocubes on ZnO surfaces, generating a ZnO-Cu$_2$O hybrid nanostructure with well-defined surface structures. In the absence of any additional sacrificial reagents, CO$_2$ reduction occurs in neutral carbonated water using the colloidal ZnO-Cu$_2$O catalyst. The resulting CH$_4$ production rate is 1080 μmol g$_{cat}^{−1}$ h$^{−1}$, which is one of the highest activities reported thus far in an aqueous med-ium. The estimated quantum efficiency (QE) is 1.5%, and the selectivity of CH$_4$ production exceeds 99%, whereas a control experiment with a TiO$_2$-Cu$_2$O catalyst mainly generates H$_2$.

## Results

**Synthesis and characterization of ZnO-Cu$_2$O hybrid nano-particles.** ZnO-Cu$_2$O hybrid nanoparticles were synthesized via a two-step process in a single batch. ZnO spheres were formed through a polyol process in the presence of PVP (poly(vinyl pyrrolidone)) behaving as a surfactant. After the complete hydrolysis of Zn precursors, a Cu precursor solution was added in situ to the reaction mixture and heated for an additional 5 min. Rapid cooling and separation yielded ZnO-Cu$_2$O nanoparticles

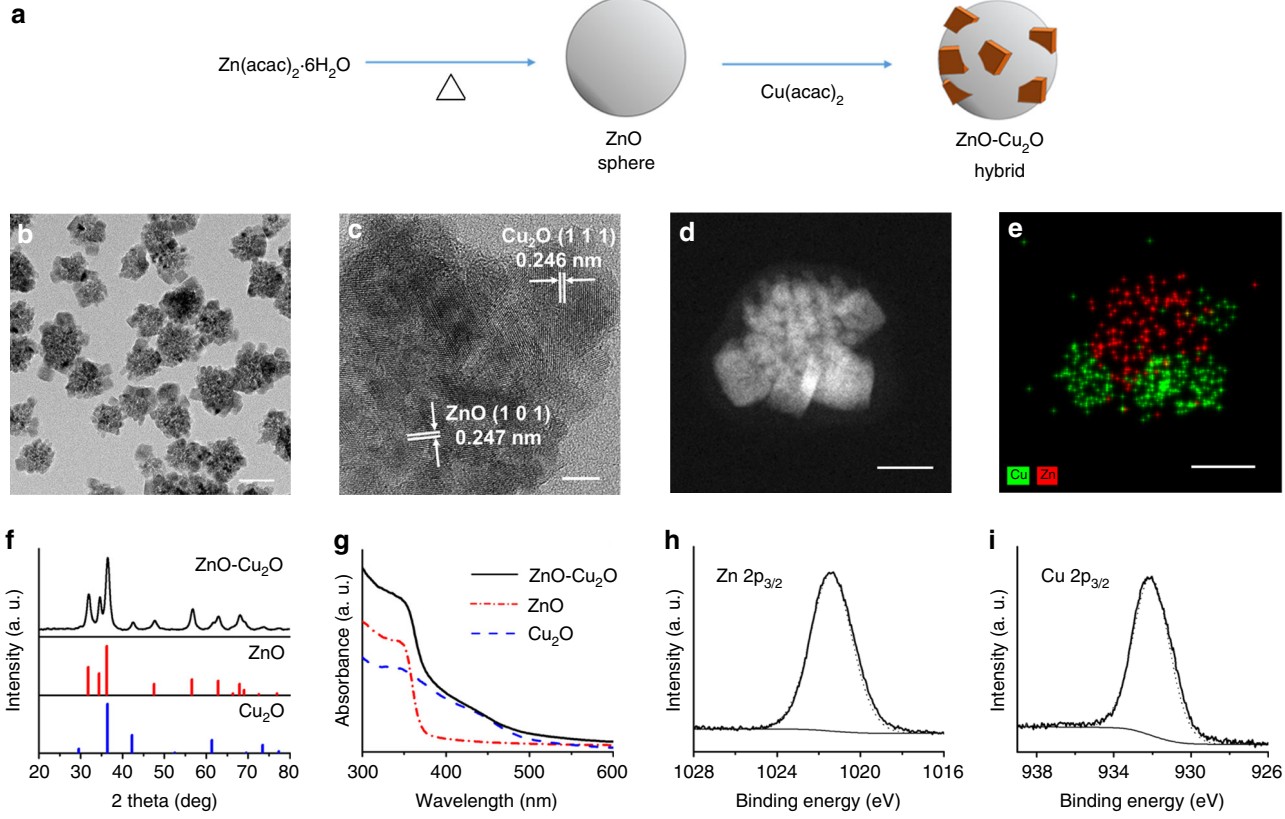

**Fig. 1** Synthesis and characterization of the ZnO-Cu$_2$O hybrid nanoparticles. **a** Synthesis of ZnO-Cu$_2$O hybrid nanoparticles via a two-step in situ process. **b** TEM image of ZnO-Cu$_2$O nanoparticles. **c** HRTEM and **d** STEM images, and **e** elemental mapping of an individual ZnO-Cu$_2$O nanoparticle. The bars represent **b** 50, **c** 5, and **d**, **e** 20 nm. **f** XRD and **g** UV–Vis spectra of ZnO-Cu$_2$O hybrid nanoparticles. XPS spectra of ZnO-Cu$_2$O hybrid nanoparticles in the regions of **h** Zn 2p$_{3/2}$ and **i** Cu 2p$_{3/2}$

(Fig. 1a)[22]. The transmission electron microscopy (TEM) image in Fig. 1b shows that each particle has an isolated structure containing multiple cubic shapes attached to a spherical core. The average diameter of the spherical cores was $40 \pm 7$ nm, and that of the cubic domains $18 \pm 3$ nm. The high-resolution TEM (HRTEM) image in Fig. 1c shows that the spherical core is actually an aggregate of small single-crystalline domains, in which the average size of each domain is estimated to be $7 \pm 1$ nm. A cubic domain attached to the core is also single crystalline. The distances between adjacent lattice fringe images are nearly identical over all domains, 0.247 nm in the core and 0.246 nm in the cubic domain. The scanning transmission electron microscopy (STEM) image in Fig. 1d clearly shows that an individual particle is composed of a spherical aggregate of small particulates in the core, with multiple rectangular domains bound to it. The elemental mapping image by energy dispersive X-ray spectroscopy (EDX) in Fig. 1e indicates that Zn and Cu components are completely separated, with Zn located in the spherical core and Cu in the cubic domains.

X-ray diffraction (XRD) data in Fig. 1f show that the pattern is an exact sum of the diffractions from hexagonal wurtzite ZnO (red, JCPDS #36-1451) and primitive cubic $Cu_2O$ (blue, JCPDS #77-0199). The single-crystalline domain size of the ZnO cores is estimated to be 7.9 nm from the FWHM of ZnO(101) peak using the Scherrer equation, in good agreement with the size measured by the HRTEM image. The ultraviolet (UV)–visible (Vis) spectrum of the ZnO-$Cu_2O$ nanoparticles in Fig. 1g is also a linear combination of those for ZnO and $Cu_2O$ nanoparticles. The band gap energies of the ZnO and $Cu_2O$ domains are 3.3 eV and 2.3 eV, respectively, estimated using Tauc plots of the UV–Vis spectra (Supplementary Fig. 1)[23]. X-ray photoelectron spectroscopy (XPS) in the Zn $2p_{3/2}$ region shows that a single peak at 1021.4 eV is assignable to Zn(II) (Fig. 1h). In particular, the spectrum in the Cu $2p_{3/2}$ region shows a single symmetric peak at 932.1 eV, indicating that there was no formation of Cu(II) during the synthesis (Fig. 1i). These observations confirm that the product is ZnO-$Cu_2O$ hybrid nanoparticles with ZnO in the cores and $Cu_2O$ in the cubic domains.

It is known that the Zn precursors were hydrolyzed in an alcoholic medium to generate Zn alcoxides, which were transformed into ZnO nanocrystalline seeds by dehydration at high temperature. The small seeds were simultaneously aggregated to yield large spheres via an oriented attachment mechanism[24]. Then, the Cu precursors were hydrolyzed and reduced to $Cu^+$ and formed $Cu_2O$ on the ZnO surface. It is noted that the distance (0.247 nm) of adjacent lattice fringe images in the sphere in Fig. 1c matches the distance of ZnO(101) planes. It is nearly identical to the distance of lattice fringes in the cubic domain of 0.246 nm, assignable to the distance of $Cu_2O$(111). This low lattice mismatch may lead to the direct growth of $Cu_2O$ on the ZnO surface forming good junctions. The crystal structure of $Cu_2O$ is primitive cubic so the $Cu_2O$ domains grow to generate cubic-type morphology by preferential adsorption of PVP on the $Cu_2O$(100) surface.

### Photocatalytic $CO_2$ conversion using ZnO-$Cu_2O$ hybrid catalysts.

The photocatalytic $CO_2$ conversion reaction was conducted using our well-defined ZnO-$Cu_2O$ hybrid nanoparticles in an aqueous medium. ZnO is an n-type metal oxide semiconductor with a large band gap (3.2 to 3.3 eV) with a low dielectric constant and high electron mobility, compared to those of $TiO_2$[25, 26]. For dye degradation reactions, ZnO catalysts show better photocatalytic activity than $TiO_2$ counterparts under the irradiation of UV–Vis light[27]. A few examples of ZnO-$Cu_2O$ heterostructures exhibited enhancement of dye degradation through the formation of p-n junctions[18, 19, 28]. In the present experiments, the pH of the

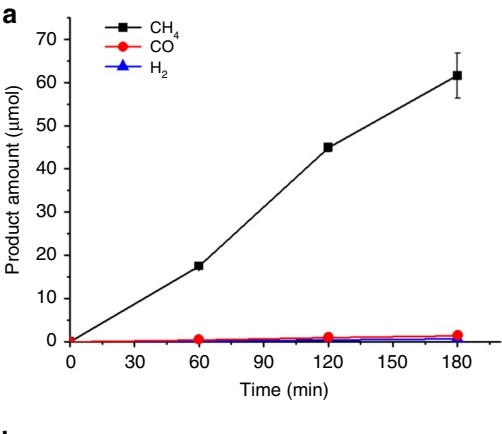

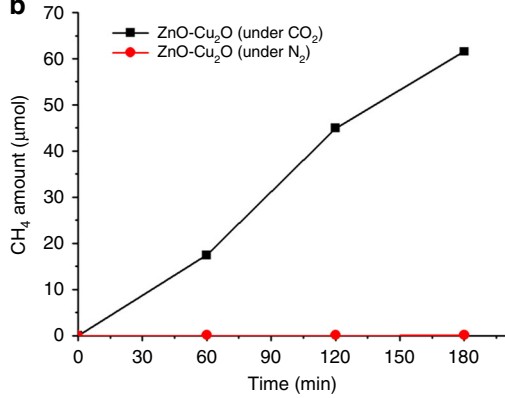

**Fig. 2** Photocatalytic $CO_2$ conversion experiments. **a** Amounts of $CH_4$ (black), CO (red), and $H_2$ (blue) production, and **b** amounts of $CH_4$ production under $CO_2$ saturation (black) and $N_2$ bubbling (red) conditions using the ZnO-$Cu_2O$ catalysts as a function of the irradiation time. The reaction conditions (**a**, **b**) were catalyst amount 19 mg, pH = 7.4, and $\lambda >$ 200 nm. The error bars were obtained from three independent experiments

reaction medium was fixed to 7.4 by the addition of perchloric acid. At this pH, the ZnO-$Cu_2O$ hybrid catalysts were stable to assess the photocatalytic reactions by prolonged UV–Vis irradiation. The $CO_2$ saturation in the aqueous medium was achieved using a 0.2 M $Na_2CO_3$ solution stirred under $CO_2$ pressure of 2.6 bar for 40 min[29]. After release of the pressure, $CO_2$ bubbling was continued at ambient pressure and temperature. By the irradiation of light using a 300 W Xe lamp, only two chemicals, $CH_4$ and CO, were detected as gaseous products using gas chromatography. $CH_4$ was the primary product with the amount of 62 μmol for 3 h (Fig. 2a), equating to catalytic activity of 1080 μmol $g_{cat}^{-1}$ $h^{-1}$ with respect to the total amount (19 mg) of the ZnO-$Cu_2O$ catalyst used in this reaction. Remarkably, the amounts of CO and $H_2$ generation were only 1.4 and 0.7 μmol, respectively, which means that the selectivity of $CH_4$ production was higher than 99%. To prove the $CH_4$ production was not from organic residues, a control experiment was carried out under $N_2$ atmosphere. By irradiation with light for 3 h, the catalyst showed a negligible $CH_4$ production of $8.4 \times 10^{-3}$ μmol (red, Fig. 2b). No reaction occurred in the absence of irradiation or catalyst, meaning that the $CH_4$ production actually originated from photocatalytic $CO_2$ reduction in the presence of the ZnO-$Cu_2O$ catalyst. The reaction in the presence of $^{13}CO_2$ was also carried out (Supplementary Fig. 2). Based on a signal at $m/e = 17$, assignable to the $^{13}CH_4$ peak in the gas chromatography–mass spectrometry (GC–MS) chromatogram when $^{13}CO_2$ and $Na_2^{13}CO_3$ were used, the percentage of $CH_4$ directly generated from $CO_2$ was estimated to be 88% during the reaction.

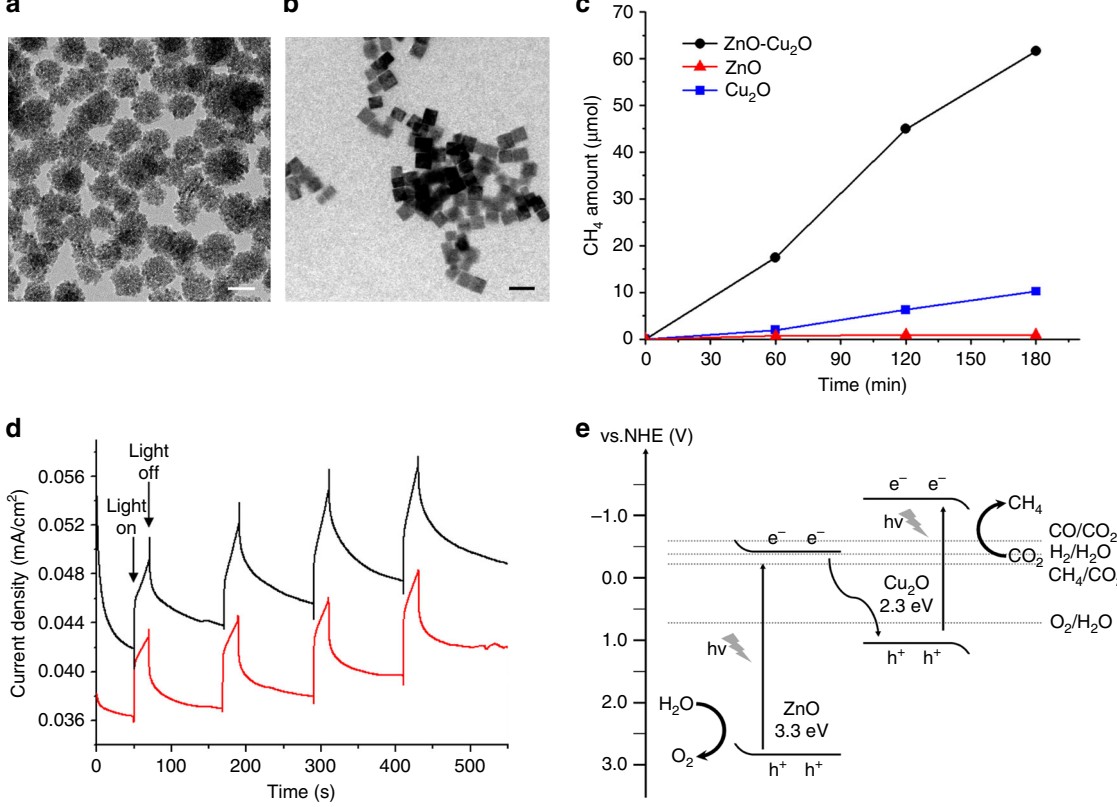

**Fig. 3** Control experiments with the ZnO and $Cu_2O$ nanoparticles and mechanistic study of the ZnO-$Cu_2O$ catalysts. TEM images of **a** ZnO spheres and **b** $Cu_2O$ nanocubes. The scale bars represent 50 nm. **c** Amount of $CH_4$ production using ZnO-$Cu_2O$ (black) catalysts, and ZnO (red) and $Cu_2O$ (blue) nanoparticles under the conditions with catalyst amount fixed to 19 mg, pH = 7.4, and $\lambda > 200$ nm. **d** Photoresponse data of the ZnO-$Cu_2O$ catalyst deposited on a FTO electrode at a potential of −0.45 V vs. Ag/AgCl in a phosphate buffer by the irradiation of UV–visible (black) and visible (red) light using a cutoff filter ($\lambda > 425$ nm). **e** Band alignment and proposed electron transfer mechanism of the ZnO-$Cu_2O$ hybrid catalysts

Eq. 1 was used to estimate the QE of $CO_2$ photoconversion to $CH_4$:[30]

$$QE(\%) = \frac{8 \times \text{Number of } CH_4 \text{ molecules}}{\text{Number of incident photons}} \times 100 \quad (1)$$

It is noted that eight electrons are required for the production of one $CH_4$ molecule from $CO_2$. The number of photons was calculated using the wavelength region between 200 to 540 nm based on the UV–Vis absorption of the catalysts (Fig. 1g) and the intensity of the incident light. The QE from photons to $CH_4$ molecules was estimated to be 1.5%.

To ensure that the ZnO-$Cu_2O$ hybrid structure is critical in $CH_4$ production, ZnO spheres and $Cu_2O$ cubes with similar size and morphology were prepared (Fig. 3a, b) and employed for photocatalytic $CO_2$ conversion. Under the experimental conditions, the activity of the ZnO spheres was estimated to be 15 $\mu mol\,g_{cat}^{-1}\,h^{-1}$, and that of the $Cu_2O$ particles was 180 $\mu mol\,g_{cat}^{-1}\,h^{-1}$, for $CH_4$ production (Fig. 3c). The lifetime of photogenerated electrons was directly measured by means of time-correlated single photon counting (TCSPC). The decay of transient absorption was measured at 620 nm, which corresponds to transition between the energy bands at the interface[31]. The photoexcited electron lifetime of ZnO-$Cu_2O$ nanoparticles ($\tau_{1/2} = 837.1$ ps) is large, compared to those of ZnO ($\tau_{1/2} = 491.4$ ps), and $Cu_2O$ ($\tau_{1/2} = 206.5$ ps) nanoparticles (Supplementary Fig. 3). This may be attributed to the interfacial states trapping electrons, which reduces the rate of charge recombination from the conduction band to the valence band of ZnO and facilitate

tunneling to the valence band of $Cu_2O$[32]. The photo-response of the ZnO-$Cu_2O$ catalysts was measured during irradiation with UV–Vis and visible light (>425 nm). The catalyst deposited on a FTO electrode generated cathodic photocurrents at an applied potential of −0.45 V vs. Ag/AgCl in a phosphate buffer, implying a p-type characteristic of the $Cu_2O$ domains. The electrode generated almost identical photocurrents under UV–Vis and visible light irradiation, indicating that the visible light absorption in the $Cu_2O$ domains is critical to generate photoelectrons (Fig. 3d).

The $CH_4$ production rates were also measured using ZnO-$Cu_2O$ catalysts synthesized from various ratios of the Zn/Cu precursors, but the activities were inferior to that of the optimized catalyst (Supplementary Fig. 4). This is because either the $Cu_2O$ domains were not fully grown on the ZnO surface, or the resulting catalyst was not uniform in its morphology. This indicates that the catalyst structure is an essential factor to maximize the catalytic performances.

**Mechanistic aspects of $CO_2$ conversion reactions.** A mechanism of $CO_2$ reduction is proposed based on these experimental results. The ZnO-$Cu_2O$ hybrid catalyst absorbs both UV and visible light corresponding to the bandgaps of 3.3 eV for the ZnO and 2.3 eV for $Cu_2O$ domains (Fig. 1g). Well-defined domain structures are expected to induce an appropriate bandgap alignment as depicted in Fig. 3e[4, 33–35]. In the Z-scheme mechanism, the effective electron transfer from the conduction band of ZnO to the valence band of $Cu_2O$ domains leads to long-lived charge separation states with the excited electrons at the conduction band of the

$Cu_2O$ domain and the holes at the valence band of the ZnO domain. The excited electrons are eventually transferred to the surface-adsorbed $CO_2$, and the holes are transferred to water molecules. With this mechanism, high activity of the ZnO-$Cu_2O$ catalyst system can be explained by the following aspects. First, ZnO has a lower dielectric constant and a higher electron mobility than $TiO_2$[25–27], which causes a low electron-hole recombination rate in photochemical reactions. The valence band edge energy of ZnO (2.8 eV vs. NHE) is far lower than the water oxidation potential (0.82 V vs. NHE), which overcomes the large over-potential commonly required for water oxidation reactions. $Cu_2O$ is also superior to CuO for $CO_2$ reduction by water, due to its large bandgap (2.4 eV) with a high-conduction band edge energy (−1.4 V vs. NHE) compared to that of CuO (−0.8 V vs. NHE)[4, 36]. It is also significantly higher than the reduction potentials of $CO_2$ to other reduced products (−0.24 ~ −0.58 V vs. NHE)[5, 9], supplying a sufficient amount of energy to the reactants. These bandgap energies render the combination of ZnO-$Cu_2O$ a good fit with the ideal band diagram for facile $CO_2$ reduction (Fig. 3e). Second, the formation of uniform domain structures facilitates electron and hole transfers to the reagents. When a photocatalyst is immersed in water, charge transfer occurs at the semiconductor-solution interface due to the equilibration of electron density between two phases[37, 38]. The net result is the formation of an electrical field at the semiconductor surface. In the case of n-type semiconductors (ZnO), when photogenerated electron-hole pairs form in the space charge region, this leads to hole transfer to the surface and water oxidation. Similarly, photogenerated electrons move to the surface and reduce $CO_2$ in p-type semiconductors ($Cu_2O$). In general, surface defects result in the formation of defect energy levels, and trap the charges, lowering the quantum yields[39]. In our ZnO-$Cu_2O$ hybrid nanoparticles, the cubic $Cu_2O$ domains are covered by the defect-less $Cu_2O(100)$ facets, and the ZnO is composed of single-crystalline domains as large as 8 nm in diameter. These have fewer surface defects than any other Zn-Cu structures[18, 19, 28], and this enhances charge transfer to the reagents. Third, the discrete morphology of the nanoparticles, by which a colloidal dispersion is readily formed in aqueous medium, is advantageous in terms of higher surface area than those of large powders or aggregates. $CO_2$ molecules should be continuously adsorbed onto the surface sites of the $Cu_2O$ domains, and protons in water also approach the reaction sites. Therefore, the high surface area resulting from the colloidal morphology is critical for the absorption of both reactants needed to achieve high activity.

The other mechanism, double charge transfer, which includes electron transfer from the conduction band of $Cu_2O$ to ZnO domains and hole transfer from the valance band of ZnO to $Cu_2O$, has also been proposed in several photoreduction systems[16, 31, 39]. However, in our catalysts, the $CH_4$ production of the pure ZnO aggregates was negligible, while the pure $Cu_2O$ nanoparticles showed a significant activity (Fig. 3c), indicating that the $Cu_2O$ domains are main active sites for $CO_2$ reduction. In the aspect of band edge energies, the Z-scheme mechanism in Fig. 3e is more reasonable to provide large overpotentials of both $CO_2$ reduction and water oxidation reactions, which the double charge transfer mechanism cannot offer. To suggest the proper photophysical mechanism, the reaction was carried out by irradiation with visible light (UV cutoff filter $\lambda > 420$ nm) under the present conditions. The $CH_4$ production was almost negligible during the reaction, and the surface state of the catalyst was unchanged after the reaction. After the removal of the cutoff filter, $CH_4$ was generated with activity similar to that of the original experiment at a fixed light intensity of 0.59 $Wcm^{-2}$ (Supplementary Fig. 5). This result indicates that the excitation of electrons in the ZnO domain is critical to activate the catalyst, and a Z-scheme is a more reliable reaction mechanism for our catalytic system.

For the issue of selectivity, this photocatalytic system provides sufficient energy, due to the Z-scheme, to provide photoexcited electrons at a high energy level for $CO_2$ reduction. It is known that the products are highly dependent upon the relative energy levels of intermediates[4, 5, 40]. Gattrell and many other researchers suggested that the radical anion of $CO_2$ is adsorbed on the metal surface and forms a carboxylic radical, which converts to CO by the interaction with surface hydrogen radical[8, 41, 42]. According to the calculations, the rate determining step of the process is the hydrogenation of CO into the formyl radical, which strongly influences the product distribution. Cu has a strong binding strength for adsorbed intermediates and facilitates the hydro-genation. More specifically, it is reported that the intermediates are particularly stabilized on the $Cu_2O(100)$ surface[43], which prevents the desorption of CO and allows efficient coupling with protons during the reaction. In the present reaction conditions, the reaction medium contains a high proton concentration at neutral pH and behaves as a rich hydrogen source that directly supplies protons[8, 41]. The resulting intermediates, such as formyl radicals or carbenes, are further hydrogenized to eventually produce $CH_4$. To understand the reaction mechanism in detail further studies are required.

The counter reaction, oxidation, should be driven by the photogenerated holes at the same time. Mostly the holes were transferred to water molecules and led to oxygen evolution, which was detectable by GC, but the PVP adsorbed on the catalyst surface might also behave as a hole scavenger during the early stage of the photocatalytic reaction.

**Comparison to the $TiO_2(P25)$-$Cu_2O$ hybrid catalysts**. For comparison, we synthesized a $TiO_2(P25)$-$Cu_2O$ hybrid structure to investigate the composition and morphology effects vs. catalytic performance. The $TiO_2(P25)$-$Cu_2O$ hybrids were synthesized via the reduction of the Cu precursors in the presence of commercial P25. A TEM image and EDX analysis indicate that

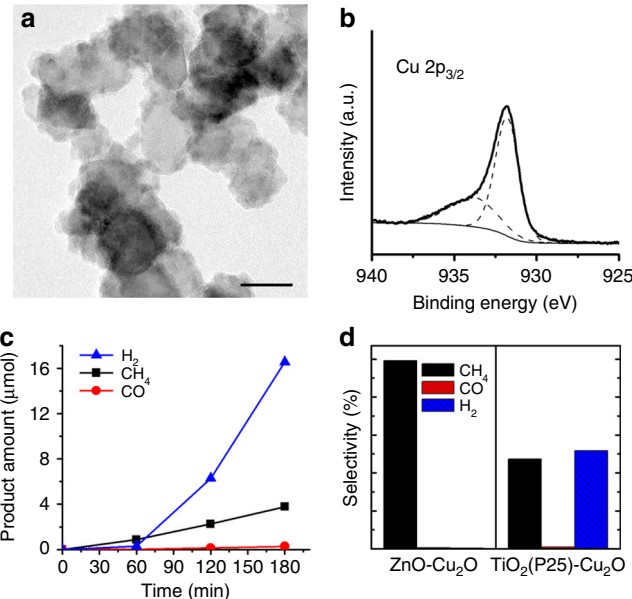

**Fig. 4** Comparison to the $TiO_2(P25)$-$Cu_2O$ hybrid catalysts. **a** TEM image of the $TiO_2(P25)$-$Cu_2O$ hybrid structure. The scale bar represents 20 nm. **b** XPS spectrum of the $TiO_2(P25)$-$Cu_2O$ hybrid structure in the region of Cu $2p_{3/2}$. **c** Amounts of $H_2$ (blue triangle), $CH_4$ (black square), and CO (red circle) production using the $TiO_2(P25)$-$Cu_2O$ catalysts as a function of the irradiation time. **d** Selectivity of gas products using ZnO-$Cu_2O$ (left) and $TiO_2(P25)$-$Cu_2O$ (right) catalysts

the Cu domains were successfully deposited on P25 (Fig. 4a and Supplementary Fig. 6). The XPS data in the region of Cu $2p_{3/2}$ also indicates the presence of Cu species on the surface (Fig. 4b). Under the present reaction conditions of $CO_2$ reduction by irradiation for 3 h, the quantity of gaseous products using the $TiO_2$(P25)-$Cu_2O$ catalysts (9.5 mg) were 3.8, 0.28, and 17 µmol for $CH_4$, CO, and $H_2$, respectively. The activity for each product was estimated as 130, 10, and 580 µmol $g_{cat}^{-1}$ $h^{-1}$ for $CH_4$, CO, and $H_2$, respectively, of which the total gas production was inferior to that using the ZnO-$Cu_2O$ catalyst (Fig. 4c). In particular, the $TiO_2$(P25)-$Cu_2O$ catalysts generated $H_2$ as a major product and $CH_4$ as the second, whereas the ZnO-$Cu_2O$ catalysts showed 99% selectivity for $CH_4$ (Fig. 4d). Regarding direct $CO_2$ conversion, the ZnO-$Cu_2O$ catalysts are superior to $TiO_2$(P25)-$Cu_2O$ for both reaction activity and selectivity.

**Stability of the ZnO-$Cu_2O$ catalyst.** The durability of the ZnO-$Cu_2O$ catalyst was tested under the present reaction conditions. The $CH_4$ production rate was constant under prolonged irradiation up to 8 h, and then dropped at over 11 h. (Fig. 5a). The reaction was carried out in a closed chamber; therefore, $CO_2$ depletion in the reaction medium may be the main reason for this activity decrease (See Supplementary Information). To prove the catalyst stability, multiple reactions with repeated $CO_2$ charging in the chamber were attempted. The reaction profile indicates that the $CH_4$ production linearly increased for more than 4 h. At this period, the reaction was stopped, the catalyst particles were re-dispersed in a fresh reaction medium with 0.2 M $Na_2CO_3$, and additional reactions were carried out under identical conditions. This process was repeated one more time. In each trial, the amount of $CH_4$ production linearly increased, and the reaction activity was nearly unchanged as shown in Fig. 5b. This implies that the catalyst stability was maintained over the reaction period of 12 h, when the fresh reaction medium was supplied. Instead of using the static reaction conditions inside the chamber, a continuous $CO_2$ flow through the reaction mixture is a potential solution to enhance the catalyst stability.

**Comparison to other photocatalysts used for $CO_2$ conversion.** The catalytic performance of the ZnO-$Cu_2O$ hybrid catalyst is listed with those of other catalysts reported in the literature (Table 1). It is very difficult to provide a direct comparison to other $CO_2$ reduction catalysts, due to different experimental conditions such as light source, reaction medium, and distinct products. However, the ZnO-$Cu_2O$ catalyst exhibits one of the highest activities and quantum yields among catalysts in aqueous media; sometimes two or three orders of magnitude higher than those of the other catalysts. The activity of the ZnO-$Cu_2O$ catalyst is comparable to the highest activities observed among solid catalysts under the conditions of high pressure $CO_2$ and high temperature.

**Discussion**

ZnO-$Cu_2O$ hybrid nanoparticles were synthesized through the direct surface growth of $Cu_2O$ on ZnO spheres. The resulting nanoparticles have ZnO and $Cu_2O$ domains with few surface defects and well-defined junctions. Photochemical $CO_2$ reduction reactions were carried out using the ZnO-$Cu_2O$ catalyst in an aqueous medium under ambient conditions. The catalyst exhibited a high reactivity of 1080 µmol $g_{cat}^{-1}$ $h^{-1}$ with a QE of 1.5% and 99% selectivity for $CH_4$. This performance for selective $CH_4$ generation is attributed to the energetic match between the ZnO and $Cu_2O$ components, and their defect-less surface and junctions. These properties suppress charge recombination and enhance effective charge transfer. This strategy to design and

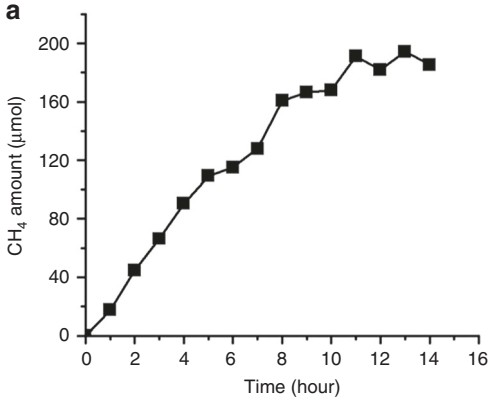

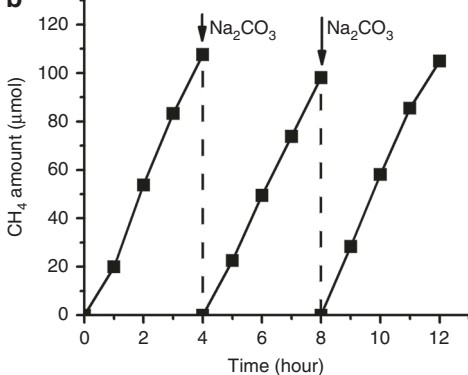

**Fig. 5** Stability experiment of the ZnO-$Cu_2O$ catalysts. **a** Amount of $CH_4$ production using the ZnO-$Cu_2O$ catalysts as a function of the irradiation time up to 14 h. The reaction conditions were pH = 7.4 and $\lambda$ > 200 nm. The $CH_4$ amount was converted based on the catalyst amount fixed to 19 mg. **b** The amount of $CH_4$ production under the identical reaction conditions except the change of the reaction medium at each 4 h reaction time

**Table 1 Comparison of the reaction conditions and performances with other catalysts for photocatalytic $CO_2$ reduction**

| Catalyst | Light source | Reaction medium | Products | Activity | Reference |
|---|---|---|---|---|---|
| ZnO-$Cu_2O$ | 300 W Xe lamp | Saturated $CO_2$ in water, 0.2 M $Na_2CO_3$ | $CH_4$ | 1080 µmol $g_{cat}^{-1}$ $h^{-1}$ QE = 1.5% | This work |
| Colloidal CdS | Medium pressure arc lamp | Saturated $CO_2$ in water, 0.1 M TMACl, 0.01 M hydroquinone | HCOOH, $CH_2O$, glyoxylic acid | 3.21 µmol $g_{cat}^{-1}$ $h^{-1}$ for HCOOH QE = 0.125% | Grimshaw et al. (ref. [44]) |
| NiO-$InTaO_4$ | Circular fluorescent lamp | Saturated $CO_2$ in water, 0.2 M NaOH | $CH_3OH$ | 2.8 µmol $g_{cat}^{-1}$ $h^{-1}$ QE = 0.0045% | Wu et al. (ref. [45]) |
| 2.0% Cu/$TiO_2$ | 8 W UV Hg lamp | Saturated $CO_2$ in water, 0.2 M NaOH | $CH_3OH$ | 19.75 µmol $g_{cat}^{-1}$ $h^{-1}$ QE = 10.02% | Wu et al. (ref. [46]) |
| Nafion/Pd-$TiO_2$ | 300 W Xe lamp | Saturated $CO_2$ in water, 0.2 M $Na_2CO_3$ | $CH_4$ | 45 µmol $g_{cat}^{-1}$ $h^{-1}$ | Choi et al. (ref. [29]) |
| Ru(bpz)$_3^{2+}$/Ru | $\lambda$ > 420 nm | Saturated $CO_2$ in water/$CH_3CH_2OH$, 0.05 M $NaHCO_3$, 0.17 M TEOA | $CH_4$ | QE = 0.04% | Willner et al. (ref. [47]) |
| Pt-$TiO_2$ thin film | 400 W Xe lamp | $CO_2$ and water flow of 3 mL $min^{-1}$ | $CH_4$ | 1361 µmol $g_{cat}^{-1}$ $h^{-1}$ QE = 2.6% | Biswas et al. (ref. [11]) |
| AuCu-P25 | 1000 W Xe lamp (AM 1.5) | 1.7 atm water saturated $CO_2$, 60 °C | $CH_4$ | 2200 µmol $g_{cat}^{-1}$ $h^{-1}$ | Garcia et al. (ref. [48]) |

synthesize well-defined nanostructures as colloidal forms could be expandable to other materials for photochemical reactions. It might also have a significant impact on the understanding of the mechanisms and key factors needed to achieve maximum catalytic performance.

## Methods

**Chemicals**. Zinc(II) acetylacetonate hexahydrate ($Zn(acac)_2 \cdot 6H_2O$, 99.995%), 1,5-pentanediol (1,5-PD, 96%), poly(vinyl pyrrolidone) (PVP, $M_w = 55,000$), copper (II) acetylacetonate ($Cu(acac)_2$, ≥ 99.95%), titanium (IV) oxide (P25, $TiO_2$, 99.5%), sodium carbonate ($Na_2CO_3$, ≥ 99.0%), and perchloric acid ($HClO_4$, 60%) were purchased from Sigma-Aldrich and used without further purification.

**Synthesis of ZnO-Cu₂O hybrid nanoparticles**. Zinc acetylacetonate hexahydrate (0.10 g, 0.40 mmol) and PVP (1.0 g, 9.0 mmol) were dissolved in 1,5-PD (50 mL) under inert conditions at 130 °C to ensure complete dissolution. The solution was heated to 225 °C for 6 min and allowed to stir for 3 min at the same temperature. Copper acetylacetonate (0.10 g, 0.40 mmol) was dissolved in 1,5-PD (5.0 mL) under an inert condition. The Cu precursor solution was added to the reaction mixture at 225 °C, followed by stirring for 5 min at the same temperature. After rapid cooling to room temperature using an ice bath, the product was separated by the addition of ethanol (60 mL) with the aid of centrifugation at 10,000 rpm. The precipitates were thoroughly washed with ethanol.

**Synthesis of ZnO spheres**. Zinc acetylacetonate hexahydrate (0.10 g, 0.40 mmol) and PVP (1.0 g, 9.0 mmol) were dissolved in 1,5-PD (50 mL) under an inert condition at 130 °C to ensure complete dissolution. The solution was heated to 225 °C for 6 min and allowed to stir for 5 min at the same temperature. After rapid cooling to room temperature using an ice bath, the product was separated by the addition of ethanol (60 mL) with the aid of centrifugation at 10,000 rpm. The precipitates were thoroughly washed with ethanol.

**Synthesis of Cu₂O nanocubes**. PVP (1.0 g, 9.0 mmol) was dissolved in 1,5-PD (50 mL) under an inert condition at 130 °C to ensure the complete dissolution. Copper acetylacetonate (0.10 g, 0.40 mmol) was dissolved in 1,5-PD (5.0 mL) under inert conditions. This Cu precursor solution was added to the reaction mixture at 225 °C, followed by stirring for 5 min at the same temperature. After rapid cooling to room temperature using an ice bath, the product was separated by the addition of ethanol (60 mL) with the aid of centrifugation at 10,000 rpm. The precipitates were thoroughly washed with ethanol.

**Synthesis of TiO₂(P25)-Cu₂O hybrid nanoparticles**. Titanium (IV) oxide (P25, 0.030 g, 0.40 mmol) and PVP (0.5 g, 4.5 mmol) were dissolved in 1,5-PD (50 mL) under inert conditions at 130 °C to ensure complete dissolution. Copper acetylacetonate (0.10 g, 0.40 mmol) was dissolved in 1,5-PD (5.0 mL) under an inert condition. The Cu precursor solution was added to the reaction mixture at 225 °C, followed by stirring for 5 min at the same temperature. After rapid cooling to room temperature using an ice bath, the product was separated by the addition of ethanol (60 mL) with the aid of centrifugation at 10,000 rpm. The precipitates were thoroughly washed with ethanol.

**Characterization**. TEM images and energy dispersive X-ray diffraction (EDX) data of the nanoparticles were obtained by FEI Tecnai G2 F30 S-Twin (300 kV, KAIST), and HRTEM images and elemental mapping were obtained by FEI Titan cubed G2 60–300 (double Cs corrected, KAIST) transmission electron microscopes. Samples were prepared by dropping a few samples dispersed in ethanol on carbon-coated 200 mesh nickel grids (Ted Pella Inc.). XRD patterns of the samples were recorded on a Rigaku D/MAX-2500 diffractometer. X-ray photoelectron spectra (XPS) were obtained by K-alpha X-ray photoelectron spectroscopy (Thermo VG Scientific). UV–Vis spectra were measured on a UV-3600 UV-vis-NIR spectrophotometer (Dong-il Shimadzu Corp.). TCSPC was measured by a FL920 spectrometer (Edinburgh Instruments).

**Photocatalytic CO₂ conversion experiments**. The ZnO-Cu₂O catalysts (19 mg) were dispersed in a 0.2 M $Na_2CO_3$ aqueous solution (20 mL), and the dispersion was neutralized to pH = 7.4 by the addition of $HClO_4$. The reactor was a home-made quartz flask with a total volume of 41 mL. Supercritical-fluid grade $CO_2$ gas was used to avoid any hydrocarbon contamination. To reach $CO_2$ saturation in the reaction medium, the catalyst dispersion was stirred for 40 min in a high pressure chamber under a $CO_2$ pressure of 2.6 bar. After the pressure release, $CO_2$ gas was transferred to the quartz reactor and was additionally bubbled at ambient pressure and temperature for 30 min. Photocatalytic $CO_2$ conversion was conducted by irradiation from a Xe lamp (300 W, Oriel) equipped with a 10 cm IR water filter. During the reaction, the gas product was collected using a needle-type probe passing through a sealed rubber septum. The gas samples were analyzed by thermal conductivity detector (TCD) and flame ionization detector (FID) equipped with a carboxen 1000 column (Supelco) via gas chromatography (YL6100 GC). Before the

FID detector, a methanizer (500 mg Ni, ~65 wt% on silica/alumina (Agilent)) was equipped for the detection of CO and $CO_2$. To avoid the oxidation of the methanizer, a valve was connected and adjusted by a program for the ventilation of evolved oxygen. For the isotope study, the gas samples were analyzed by GC–MS (Agilent 7890 A/5977B) equipped with a HP-5MS (Agilent) capillary column.

**Data availability**. The data that support the findings of this study are available from K.M.N. (email: namkimin.chem@gmail.com) or H.S. (email: hsong@kaist.ac.kr) upon reasonable request.

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

## Acknowledgements

This work was supported by the Saudi-Aramco and KAIST $CO_2$ management center. This work was also supported by IBS-R004-D1 and the National Research Foundation of Korea (NRF) funded by the Korea Government (MSIP) (NRF-2015R1A2A2A01004196).

## Author contributions

K.-L.B. and J.K. contributed equally to this work. H.S. designed this study. H.S., K.M.N., K.-L.B., and J.K. wrote the manuscript. K.-L.B. conducted photocatalytic $CO_2$ reduction experiments and mechanistic study. J.K. and C.K.L. carried out catalyst synthesis and characterization.

## Additional information

**Competing interests:** The authors declare no competing financial interests.

