## [Peer Review File · Nature Communications]

Reviewer #1 (Remarks to the Author):

The manuscript by Bae et al reports an efficient photocatalytic CO₂ reduction to methane using a heterostructure composed of a central aggregate of ZnO nanocrystals surrounded by cubic Cu₂O crystals. The authors claim high selectivity towards the formation of CH₄ over the formation of CO. I find the topic to be interesting and suitable for the journal. The reported efficiency is certainly of note. The characterization is properly written, but the discussion of the results is unfortunately of lesser standard. In addition, I have serious doubts about the proposed mechanism and the accuracy of some of the measurements. At this stage, I cannot recommend publication of the manuscript. Only if a proper mechanism can be proposed and verified, as indicated below, I suggest the revised manuscript might be re-considered for publication.

1. The authors claim that the CO₂ reduction step proceeds on ZnO crystals due to Type II alignment between ZnO and Cu₂O. This would be very unusual as ZnO is not known for good photocatalytic properties towards CO₂ reduction (without the use of noble metal co-catalysts). On the other hand Cu₂O is considered to be one of the best candidates. In fact the measurements taken by the authors for isolated ZnO and Cu₂O confirm this trend (15micromol vs 180micromol). The authors would need very good arguments for the reaction to proceed on ZnO, which are not present in the current manuscript.

2. The authors ascribe the selectivity to the level of conduction band edge of ZnO which they locate between the redox level of reduction of CO₂ to CO and CH₄, but this is highly controversial. The process is largely limited by energy levels in intermediate steps, not by the overall thermodynamic considerations.

3. The authors claim that oxygen is produced in large amounts on the oxidation side, but do not provide any evidence. Again formation of oxygen by Cu₂O would be quite usual, because of very small overpotential. O₂ formation typically requires much larger overpotentials than available with Cu₂O.

4. The mix of products reported by the authors corresponds to using TCD detector in the gas chromatograph. I find it very strange that the authors report using an FID detector. CO is not really detectable by the FID detector, unless a methanizer is used on the line before the detector. This is not mentioned in the manuscript. I suspect that the authors in fact used a TCD detector, but that left them unable to detect a large number of other possible products not considered in the manuscript, such as alcohols, formaldehyde, propene and others which are commonly detected (albeit often in small quantities) among the products. In this context, I find the claimed selectivity to be not yet proven.

In summary, some of the findings are interesting, but at this stage the manuscript does not in my opinion meet the standards of publication in Nature Communications.

Reviewer #2 (Remarks to the Author):

This manuscript reports the photocatalytic activity of small particle size spherical ZnO having Cu₂O nanocubes for CO₂ reduction by H₂O. The reaction is of large interest and the data presented are in the millimol/g•h with 1.5 % quantum efficiency that is a remarkable activity. Nevertheless, publication in Nature Communications of this submission is not recommended, because the work has not addressed the main drawbacks of the two metal oxide, i.e., their instability and their tendency to undergo photocorrosion and deactivation upon extended use. Just a simple comment that the material is “stable enough” is not convincing in view of the extensive data in the literature that, both component ZnO and Cu₂O, even though quite photoactive at short times, eventually deactivate. Figure 5 that is key in this issue shows a significant decrease in the production rate from 0 to 2 h compared to 10-14 h in which the reaction does not progress. Why? Based on this stability issue, publication should not proceed.

Other points that the authors may consider are the following:

The proportion between ZnO and Cu₂O has not been optimized. Only a material with a fixed percentage has been studied.

The text comments twice that assignment of the particles in HRTEM is based on the difference between 0.247 (ZnO) and 0.246 nm (Cu₂O). What is the resolution of the instrument? In contrast to the extremely high resolution of the instrument, the claimed epitaxial growth of Cu₂O on top of ZnO has not been supported with the corresponding HRTEM images of this region.

¹³C₂O has to be used as starting material and detect the corresponding ¹³CH₄ in the required ¹³C percentage.

It is known that the binding energy value in XPS cannot differentiate between Cu(0) and Cu(I). However, the authors attribute the XPS Cu_{2p} peak exclusively to Cu(I).

Since Cu₂O is active under visible light, the photoresponse of the material has to be presented.

Figure 3 e regarding the mechanism implies that both ZnO and Cu₂O can absorb photons, leading to equal efficiency? ZnO absorbs in the UV, while Cu₂O absorbs in the visible. This issue of which component is most efficient should be addressed.

According to Figure 3 e, irradiation of ZnO under UV light should also produce the same reaction without the need of Cu₂O. This has to be confirmed.

Reviewer #3 (Remarks to the Author):

This manuscript reports an impressive result in terms of photocatalytic efficiency for the conversion of CO₂ to methane. The authors claim they have achieved this result by synthesizing an improved p-n junction between nanoparticles of ZnO and Cu₂O, and they provide strong evidence supporting this claim. However, in the manuscript they make several statements that are either incorrect or speculative (when proof could be obtained in a direct manner). These items need to be considered prior to publication:

1. Lines 130-132, The authors make a CO₂ saturated electrolyte by starting with a 0.2M NaCO₃ solution which is acidified with perchloric acid to pH=7.4 under a pressure of CO₂(g). But, at this pH carbonate is converted to CO₂. So, this procedure makes little sense. Is there something going on here that is not being described?
2. Lines 138-141, The authors carry out a run under N₂, which only produces a trace of CH₄ to “prove” that the observed CH₄ derives from CO₂. This is a necessary but insufficient control. One can image a number of reasons that methane is not produced under nitrogen, but it does not come from CO₂. This is especially true because of the comment made in point 1 about the authors “hidden” source of CO₂. The essential experiment to carry out is a run using ¹³CO₂ showing that ¹³CH₄ is formed. This manuscript should not be published without this critical control experiment.
3. Lines 152-155, This is not the normal definition of quantum yield. The QE for a product is normally the moles of product formed per moles of photons incident. By inserting a factor of 8, the authors have converted the quantum yield from CO production to the quantum yield for electron production (leading to CO formation). This becomes important when they compare their results to other works.
4. Line 196 indicates that water is oxidized in this reaction. However, no data is provided showing the formation of O₂. This needs to be demonstrated if one wants to claim water oxidation.
5. Lines 201-204, The selectivity of the process for methane is impressive, but it is not due to the indicated conduction band position.
6. Line 237-8, the authors state, “Regarding direct CO₂ conversion, the ZnO-Cu₂O catalysts are definitely superior to TiO₂-Cu₂O for both reaction activity and selectivity.” But, in fact, what the data proves is that their catalyst is an improvement over P25-Cu₂O only.
7. Line 252-256, This is a false statement. Neither the reduction of CO₂ to methane or the oxidation of water to O₂ is reversible by any stretch of the imagination, under any set of conditions present in the experiment under consideration. In fact, these reactions are often used to illustrate the prototypically irreversible reaction.

8. Line 257-258, Why do the authors believe that O₂ is oxidizing the Cu₂O phase instead of this phase being oxidized by the photogenerated holes?

Once these issues have been addressed, I believe this paper will be publishable.

For Reviewer #1:

The manuscript by Bae et al reports an efficient photocatalytic CO₂ reduction to methane using a heterostructure composed of a central aggregate of ZnO nanocrystals surrounded by cubic Cu₂O crystals. The authors claim high selectivity towards the formation of CH₄ over the formation of CO. I find the topic to be interesting and suitable for the journal. The reported efficiency is certainly of note. The characterization is properly written, but the discussion of the results is unfortunately of lesser standard. In addition, I have serious doubts about the proposed mechanism and the accuracy of some of the measurements.

Author response: We deeply appreciate the reviewer's evaluation that our work is interesting and our results are certainly of note. We agree the reviewer's concern on the proposed mechanism and its explanation. We have checked our measurements to enhance the accuracy, and have completely re-written the section of band alignment and electron transfer mechanism related to Fig. 3.

1. The authors claim that the CO₂ reduction step proceeds on ZnO crystals due to Type II alignment between ZnO and Cu₂O. This would be very unusual as ZnO is not known for good photocatalytic properties towards CO₂ reduction (without the use of noble metal co-catalysts). On the other hand Cu₂O is considered to be one of the best candidates. In fact the measurements taken by the authors for isolated ZnO and Cu₂O confirm this trend (15micromol vs 180micromol). The authors would need very good arguments for the reaction to proceed on ZnO, which are not present in the current manuscript.

Author response: We totally respect the reviewer's opinion for the description of mechanistic aspects. In fact, there have been two opposite pathways proposed for metal oxide hybrid catalytic systems; one is the electron transfer from the conduction band of one material at a higher energy level to the conduction band of the other material at a lower energy level, which was what we proposed in our manuscript. The ZnO islands on CuO nanowires (Biswas et al. ACS Appl. Mater. Interfaces 7, 5685-5692 (2015)) and Cu₂O/TiO₂ porous materials (Ye et al. Nanotechnology 25, 165402 (2014)) were the systems explained using this model of bandgap alignment. On the contrary, most of other literatures used Z-scheme-like reaction pathways where the electron is transferred from the conduction band of one material at a low energy level to the valence band of another material (Grimes et al. ACS Nano 4, 1259-1278 (2010); Li et al. ACS Appl. Mater. Interfaces 7, 8631-8639 (2015)). Although there remains some ambiguity between the opposite reaction pathways, we think that the latter mechanism is more appropriate for our catalytic system, because of the reasons that the reviewer commented. Figure 3c clearly shows that the Cu₂O sites are more active than the ZnO sites as a pure form. The sufficient energy of the excited electrons on the conduction band of the Cu₂O sites leads to the effective reduction of CO₂ into other compounds. On the other hand, the water oxidation generally requires a large overpotential, therefore, the hole transfer from the valence band of the ZnO domains is more reasonable to understand the high activity of this catalytic system. We have changed Fig. 3e, and have re-written the section of "Mechanistic aspects of CO₂ conversion reactions", by the proposition of the latter pathway for CO₂ reduction.

Revision made:

Page 10: Fig. 3e was changed into the proper diagram depicting bandgap alignment and reaction pathways.

Figure 3 | (e) Band alignment and proposed electron transfer mechanism of the ZnO-Cu₂O hybrid catalysts.

Pages 10-13: The section of “Mechanistic aspects of CO₂ conversion reactions” was completely re-written.

2. The authors ascribe the selectivity to the level of conduction band edge of ZnO which they locate between the redox level of reduction of CO₂ to CO and CH₄, but this is highly controversial. The process is largely limited by energy levels in intermediate steps, not by the overall thermodynamic considerations.

Author response: For the issue of selectivity, we also agree the reviewer’s opinion, in which the product selectivity is limited by the energies of intermediates with kinetic factors, but is not determined by thermodynamic energy profiles. We have begun to study gas-phase reaction conditions using the present ZnO-Cu₂O catalysts, and as a preliminary result, the CH₄ generation was significantly diminished with the large production of CO when the small amount of water was used. This result indicates that water behaves as a rich hydrogen source, leading to nearly quantitative production of CH₄ over the other products. It is also reported that adsorbed CO are particularly stabilized on the Cu₂O(100) surface, which may allow more efficient coupling with adsorbed protons during the reaction. We have inserted these discussions on product selectivity in the main text.

Revision made:

Pages 12-13: We inserted a paragraph as “For the issue of selectivity, this photocatalytic system provides sufficient energy based on the Z-scheme with photoexcited electrons at a high energy level to CO₂ reduction. It is known that the products are highly dependent upon the relative energy levels of intermediates. In the present reaction system, the reaction medium, water, with a high proton concentration behaves a rich hydrogen source, and leads to the production of CH₄ more effectively. In addition, it is reported that the intermediates such as CO are particularly stabilized on the Cu₂O(100) surface, which may allow more efficient coupling with adsorbed protons during the reaction.”.

3. The authors claim that oxygen is produced in large amounts on the oxidation side, but do not provide any evidence. Again formation of oxygen by Cu₂O would be quite usual, because of very small overpotential. O₂ formation typically requires much larger overpotentials than

available with Cu₂O.

Author response: As we fully reflected the reviewer's recommendation, we have changed the proposed reaction mechanism in Fig. 3e. In this scheme, the holes remain at the valence band of the ZnO domain, and oxygen forms on this ZnO surface. The valence band edge energy of ZnO (2.8 eV vs. NHE) is far lower than the water oxidation potential (0.82 V vs. NHE), which overcomes the large overpotential commonly required for water oxidation reactions. As a direct evidence, we have detected the oxygen amount increasing along the reaction progress.

Revision made:

Page 10: We inserted the sentences, "It provides effective electron transfer from the conduction band of ZnO to the valence band of Cu₂O domains, resulting in long-lived charge separation states with the excited electrons at the conduction band of the Cu₂O domain and the holes at the valence band of the ZnO domain." and "The valence band edge energy of ZnO (2.8 eV vs. NHE) is far lower than the water oxidation potential (0.82 V vs. NHE), which overcomes the large overpotential commonly required for water oxidation reactions."

Page 14: We omitted the description of oxygen formation for the discussion of the stability issue. The section for the discussion of catalyst stability was completely re-written (See below responses for Reviewer #2).

4. The mix of products reported by the authors corresponds to using TCD detector in the gas chromatograph. I find it very strange that the authors report using an FID detector. CO is not really detectable by the FID detector, unless a methanizer is used on the line before the detector. This is not mentioned in the manuscript. I suspect that the authors in fact used a TCD detector, but that left them unable to detect a large number of other possible products not considered in the manuscript, such as alcohols, formaldehyde, propene and others which are commonly detected (albeit often in small quantities) among the products. In this context, I find the claimed selectivity to be not yet proven.

Author response: We are sorry for missing the description of using both TCD and FID detectors. The methanizer (500 mg Ni, ~65 wt% on silica/alumina (Agilent)) was also equipped to the gas chromatography for the detection of CO and CO₂. We have described the details in the Method section.

Revision made:

Page 20, in Method: We added a sentence, "The gas samples were analyzed by TCD and FID detectors equipped with a carboxen 1000 column (Supelco) in gas chromatography (YL6100 GC). Before the FID detector, a methanizer (500 mg Ni, ~65 wt% on silica/alumina (Agilent)) was equipped for the detection of CO and CO₂. To avoid the oxidation of methanizer, a valve was connected and adjusted by a program for the ventilation of evolved oxygen."

For Reviewer #2:

1. This manuscript reports the photocatalytic activity of small particle size spherical ZnO having Cu₂O nanocubes for CO₂ reduction by H₂O. The reaction is of large interest and the

data presented are in the millimol/g·h with 1.5 % quantum efficiency that is a remarkable activity. Nevertheless, publication in Nature Communications of this submission is not recommended, because the work has not addressed the main drawbacks of the two metal oxide, i.e., their instability and their tendency to undergo photocorrosion and deactivation upon extended use. Just a simple comment that the material is “stable enough” is not convincing in view of the extensive data in the literature that, both component ZnO and Cu₂O, even though quite photoactive at short times, eventually deactivate. Figure 5 that is key in this issue shows a significant decrease in the production rate from 0 to 2 h compared to 10-14 h in which the reaction does not progress. Why?

Author response: At first, we appreciate the reviewer’s evaluation of the importance of our work. We also agree the reviewer’s main concern on the catalyst stability. As the reviewer commented, both components of ZnO and Cu₂O are known to be unstable under the low pH conditions. But, under the nearly neutral conditions, the particles maintained their original structure when we carried out the reactions for more than 24 h. For Fig. 5, we reported some saturation behaviors of the reactions by prolonged irradiation, which were explained by the reaction equilibrium between the reactants and products. However, as the other reviewer claimed, we were totally incorrect, because neither CO₂ reduction into methane nor water oxidation are reversible reactions. We think that the decrease of the CH₄ production at 10-14 h is mainly due to the depletion of CO₂. To prove it, we have carried out more stability tests as follows:

Experiment 1: Additional 10 h reaction after the reaction for 14 h

We have carried out the CO₂ reduction experiment for 14 h under the present reaction conditions. Then, the catalyst particles were re-dispersed in a fresh reaction medium with 0.2 M Na₂CO₃, and the reaction was carried out again under the identical conditions. The catalysts were still active after the 14 h reaction, and showed a similar and even better activity than that of the first reaction.

Experiment 2: repeating 4 h reactions three times using fresh reaction media

In this experiment, the catalyst particles were re-dispersed in the fresh reaction medium with 0.2 M Na_2CO_3 after the first 4 h reaction, and the reaction was carried out. This process was repeated again to show the catalyst stability lasting for 12 h. As shown in the figure, the catalytic activity was regenerated after the use of the fresh reaction medium.

Based on these tests, we concluded that the catalyst stability maintains during the reaction for more than 12 h, when the fresh reaction medium is supplied. The present reaction proceeded inside the closed chamber, resulting in the CO_2 depletion by the prolonged reaction time. This would be a main cause for the significant decrease of the activity at the late stage of the reaction. The CO_2 depletion may change the pH of the reaction medium and slow down the CO_2 reduction reaction, which leads to the oxidation of Cu_2O surface into CuO by photogenerated holes. We have added the data of Experiment 2 in Fig. 5b, and have rewritten the section of “Stability of the $\text{ZnO-Cu}_2\text{O}$ catalysts”.

Revision made:

Pages 14-15: The section of “Stability of the $\text{ZnO-Cu}_2\text{O}$ catalysts” was also completely re-written. Fig. 5b was changed as follows.

Figure 5 | Stability experiment of the $\text{ZnO-Cu}_2\text{O}$ catalysts. (a) using the $\text{ZnO-Cu}_2\text{O}$ catalysts as a function of the irradiation time up to 14 h. The reaction conditions were $\text{pH} = 7.4$ and $\lambda > 200$ nm. The CH_4 amount was converted based on the catalyst amount fixed to 19

mg. (b) The amount of CH₄ production under the identical reaction conditions except the change of the reaction medium at each 4 h reaction time.

2. Other points that the authors may consider are the following:

The proportion between ZnO and Cu₂O has not been optimized. Only a material with a fixed percentage has been studied.

Author response: We have already carried out the optimization of the ratio between ZnO and Cu₂O domains for the catalyst purpose. The CH₄ production rate was clearly dependent upon the Zn/Cu ratio as follows. However, in the cases of other catalysts with the Zn/Cu ratios distinct from 1 : 1, either the Cu₂O domain was not fully grown or the ZnO-Cu₂O catalyst structure was not sufficiently defined well. Our purpose for this manuscript is the importance of well-defined nanostructures for CO₂ conversion reactions, therefore, we did not include this optimization process in the main text.

3. The text comments twice that assignment of the particles in HRTEM is based on the difference between 0.247 (ZnO) and 0.246 nm (Cu₂O). What is the resolution of the instrument? In contrast to the extremely high resolution of the instrument, the claimed epitaxial growth of Cu₂O on top of ZnO has not been supported with the corresponding HRTEM images of this region.

Author response: As the reviewer commented, we have checked the HRTEM images of different samples. In fact, it is very hard to clearly see the junction area, because the ZnO domain is an aggregate of small single-crystalline domains. See below for a representative HRTEM image. We still think that the low lattice mismatch between ZnO and Cu₂O domains enables the direct attachment of the Cu₂O seeds on the surface of ZnO, but we agree that the term of “epitaxial growth” is misused, because it is generally used for good crystallographic alignment between the different domains in a large area during the thin film growth process. We have changed the descriptions related to “epitaxial”.

Revision made:

Pages 6-7: We modified the sentence as “This low lattice mismatch may lead to the direct growth of Cu_2O on the ZnO surface forming good junctions.”.

Page 12: We modified the sentence as “As shown in Fig. 1c, the average distance of $\text{Cu}_2\text{O}(111)$ planes matches that of $\text{ZnO}(101)$ planes, which forms good contact between the different domains.”.

Page 16: We modified the sentence as “ $\text{ZnO-Cu}_2\text{O}$ hybrid nanoparticles were synthesized through the direct surface growth of Cu_2O on ZnO spheres.”.

4. $^{13}\text{CO}_2$ has to be used as starting material and detect the corresponding $^{13}\text{CH}_4$ in the required ^{13}C percentage.

Author response: We have conducted an isotope experiment using $^{13}\text{CO}_2$ and $\text{Na}_2^{13}\text{CO}_3$. The gas product was analyzed by GC/MS. A signal at $m/e = 17$ increased a lot when the ^{13}C reagents were used for the reaction, indicative of the production of $^{13}\text{CH}_4$. This result implies that CH_4 was directly produced from CO_2 , which is relevant to the control experiment of no CH_4 formation under a N_2 flow.

Revision made:

Supplementary Figure S2:

Supplementary Figure 2 | GC-MS chromatogram at $m/e = 17$ using ^{13}CO and $\text{Na}_2^{13}\text{CO}_2$ as carbon sources for CO_2 reduction.

Pages 8-9: We inserted sentences as “The reaction in the presence of $^{13}\text{CO}_2$ was also carried out (Supplementary Fig. 2). A signal at $m/e = 17$, assignable to the $^{13}\text{CH}_4$ peak, increased a lot in the gas chromatography-mass spectrometry (GC-MS) chromatogram when $^{13}\text{CO}_2$ and $\text{Na}_2^{13}\text{CO}_3$ were used, which were another indication of the direct CO_2 reduction into CH_4 .”.

5. It is known that the binding energy value in XPS cannot differentiate between $\text{Cu}(0)$ and $\text{Cu}(I)$. However, the authors attribute the XPS $\text{Cu}2p$ peak exclusively to $\text{Cu}(I)$.

Author response: As the reviewer commented, $\text{Cu}(0)$ and $\text{Cu}(I)$ cannot be distinguished in the XPS spectrum. However, XPS data can indicate the absence of $\text{Cu}(II)$ species. By the combination of XRD data, it was informed that Cu_2O was exclusively generated from the reaction. We have modified the descriptions related to the XPS data.

Revision made:

Page 6: We changed the sentence as “In particular, the spectrum in the $\text{Cu } 2p_{3/2}$ region shows a single symmetric peak at 932.1 eV, indicating that there were no formation of $\text{Cu}(II)$ during the synthesis (Fig. 1i).”.

Page 14: We changed the sentence as “The XPS data in the region of $\text{Cu } 2p_{3/2}$ also indicates the presence of Cu species on the surface (Fig. 4b).”.

6. Since Cu_2O is active under visible light, the photoresponse of the material has to be presented.

Author response: As the reviewer recommended, we have checked the photoresponse of our catalyst, $\text{ZnO-Cu}_2\text{O}$, by the irradiation of UV-vis and visible light ($> 425 \text{ nm}$). The catalyst deposited on a FTO electrode generated cathodic photocurrents at an applied potential of $-0.45 \text{ V vs. Ag/AgCl}$ in a phosphate buffer, implying the p-type characteristics of the Cu_2O domains. The electrode generated almost identical photocurrents under UV-Vis and visible light irradiation, indicating that the Cu_2O is highly effective in the visible light region.

Revision made:

Page 10: We inserted the sentences as, “The photoresponse of the ZnO-Cu₂O catalysts was measured by the irradiation of UV-Vis and visible light (> 425 nm). The catalyst deposited on a FTO electrode generated cathodic photocurrents at an applied potential of -0.45 V vs. Ag/AgCl in a phosphate buffer, implying the p-type characteristics of the Cu₂O domains. The electrode generated almost identical photocurrents under UV-Vis and visible light irradiation, indicating that the Cu₂O is highly effective in the visible light region (Supplementary Fig. 4).”.

Supplementary Figure S4:

Supplementary Figure 4 | Photoresponse data of the ZnO-Cu₂O catalyst deposited on a FTO electrode at a potential of -0.45 V vs. Ag/AgCl in a phosphate buffer by the irradiation of UV-Vis (black line) and visible (red line) light using a cut-off filter (> 425 nm).

7. Figure 3e regarding the mechanism implies that both ZnO and Cu₂O can absorb photons, leading to equal efficiency? ZnO absorbs in the UV, while Cu₂O absorbs in the visible. This issue of which component is most efficient should be addressed.

8. According to Figure 3 e, irradiation of ZnO under UV light should also produce the same reaction without the need of Cu₂O. This has to be confirmed.

Author response: We agree that our explanation of the mechanistic aspect was incorrect. Figure 3c clearly shows that the Cu₂O sites are more efficient than the ZnO sites as a pure form for CO₂ reduction. Apparently, the Cu₂O sites should be the active surface where the CO₂ reduction occurs. In fact, there have been two opposite pathways proposed for metal oxide hybrid catalytic systems; one is the electron transfer from the conduction band of one material at a higher energy level to the conduction band of the other material at a lower energy level, which was what we proposed in our manuscript. The ZnO islands on CuO nanowires (Biswas et al. ACS Appl. Mater. Interfaces 7, 5685-5692 (2015)) and Cu₂O/TiO₂ porous materials (Ye et al. Nanotechnology 25, 165402 (2014)) were the systems explained using this model of bandgap alignment. On the contrary, most of other literatures used Z-scheme-like reaction pathways where the electron is transferred from the conduction band of one material at a low energy level to the valence band of another material (Grimes et al. ACS Nano 4, 1259-1278 (2010); Li et al. ACS Appl. Mater. Interfaces 7, 8631-8639 (2015)).

Although there remains some ambiguity between these opposite reaction pathways, we think that the latter mechanism is more appropriate for our catalytic system, because of the reasons that the reviewer commented well. The sufficient energy of the excited electrons on the conduction band of the Cu₂O sites leads to the effective reduction of CO₂ into other compounds. On the other hand, the water oxidation generally requires a large overpotential, therefore, the hole transfer from the valence band of the ZnO domains is more reasonable to understand the high activity of this catalytic system. We have changed Fig. 3e, and have re-written the section of “Mechanistic aspects of CO₂ conversion reactions”, by the proposition of the latter pathway for CO₂ reduction.

Revision made:

Page 10: Fig. 3e was changed into the proper diagram depicting bandgap alignment and reaction pathways.

Figure 3 | (e) Band alignment and proposed electron transfer mechanism of the ZnO-Cu₂O hybrid catalysts.

Pages 10-13: The section of “Mechanistic aspects of CO₂ conversion reactions” was completely re-written.

For Reviewer #3:

This manuscript reports an impressive result in terms of photocatalytic efficiency for the conversion of CO₂ to methane. The authors claim they have achieved this result by synthesizing an improved p-n junction between nanoparticles of ZnO and Cu₂O, and they provide strong evidence supporting this claim. However, in the manuscript they make several statements that are either incorrect or speculative (when proof could be obtained in a direct manner). These items need to be considered prior to publication. Once these issues have been addressed, I believe this paper will be publishable.

Author response: We deeply appreciate the reviewer’s evaluation of our achievement and its strong evidences. We agree that the original manuscript has several incorrect statements; therefore, we have carefully addressed all issues that the reviewer kindly commented.

1. Lines 130-132, The authors make a CO₂ saturated electrolyte by starting with a 0.2M

Na₂CO₃ solution which is acidified with perchloric acid to pH=7.4 under a pressure of CO₂(g). But, at this pH carbonate is converted to CO₂. So, this procedure makes little sense. Is there something going on here that is not being described?

Author response: We have used the right condition of pH = 7.4 as we described in the manuscript. As the reviewer commented, most of the CO₂ reduction reactions were carried out under the basic conditions dissolving more CO₂ in water as a carbonate form. However, multiple protons are particularly needed for CH₄ production; therefore, there should be a middle pH to satisfy these two opposite requirements. As a matter of fact, there are several results using neutral conditions for CO₂ reduction (Chen et al. J. Phys. Chem. Solids 73, 661-669 (2012); Chen et al. Catal. Commun. 8, 1546-1549 (2007)). In the case of AgBr/TiO₂ nanocomposites, the catalyst exhibited maximum activities for the methane and methanol production in the range of pH 6.0 – 8.5 (He et al. Catal. Today 175, 256-263 (2011)). In the Nafion/Pd-TiO₂ catalyst, even pH 1 was the best condition of the CH₄ and C₂H₆ generation compared to the conditions at pH = 3 and 11. In our experimental conditions, we think that the high proton concentration at neutral pH may help the effective formation of CH₄. We described this factor in the section for the selectivity issue.

Revision made:

Pages 12-13: We inserted the description of the proton concentration as “In the present reaction conditions, the reaction medium, water, with a high proton concentration at neutral pH behaves a rich hydrogen source, and leads to the CH₄ production more effectively. In addition, it is reported that the intermediates such as CO are particularly stabilized on the Cu₂O(100) surface, which may allow more efficient coupling with adsorbed protons during the reaction.”.

2. Lines 138-141, The authors carry out a run under N₂, which only produces a trace of CH₄ to “prove” that the observed CH₄ derives from CO₂. This is a necessary but insufficient control. One can imagine a number of reasons that methane is not produced under nitrogen, but it does not come from CO₂. This is especially true because of the comment made in point 1 about the authors “hidden” source of CO₂. The essential experiment to carry out is a run using ¹³CO₂ showing that ¹³CH₄ is formed. This manuscript should not be published without this critical control experiment.

Author response: As the reviewer recommended, we have conducted an isotope experiment using ¹³CO₂ and Na₂¹³CO₃. The gas product was analyzed by GC/MS. A signal at m/e = 17 increased a lot when the ¹³C reagents were used for the reaction, indicative of the production of ¹³CH₄. This result implies that CH₄ was directly produced from CO₂, which is relevant to the control experiment of no CH₄ formation under a N₂ flow.

Revision made:

Supplementary Figure S2:

Supplementary Figure 2 | GC-MS chromatogram at $m/e = 17$ using ^{13}CO and $\text{Na}_2^{13}\text{CO}_2$ as carbon sources for CO_2 reduction.

Pages 8-9: We inserted sentences as “The reaction in the presence of $^{13}\text{CO}_2$ was also carried out (Supplementary Fig. 2). A signal at $m/e = 17$, assignable to the $^{13}\text{CH}_4$ peak, increased a lot in the gas chromatography-mass spectrometry (GC-MS) chromatogram when $^{13}\text{CO}_2$ and $\text{Na}_2^{13}\text{CO}_3$ were used, which were another indication of the direct CO_2 reduction into CH_4 .”.

3. Lines 152-155, This is not the normal definition of quantum yield. The QE for a product is normally the moles of product formed per moles of photons incident. By inserting a factor of 8, the authors have converted the quantum yield from CO production to the quantum yield for electron production (leading to CO formation). This becomes important when they compare their results to other works.

Author response: We understand that the quantum efficiency is defined by following equation,

$$\text{QE} = \left(\sum_i M_i n_i \right) / P_m$$

where M_i is the molar concentration of component i , n_i is number of electrons participating in the synthesis of component i (e.g. $n = 8$ in the case of methane formation), and P_m is molar concentration of incident photons. Grimes et al. used this equation to compare some important photocatalytic reactions in terms of quantum efficiency (Grimes et al. ACS Nano 2010, 4, 1259-1278 (2010)). Many of other literatures used the same definition to calculate the quantum efficiency (Bocarsly et al. J. Am. Chem. Soc. 130, 6342-6344 (2008); Ye et al. Chem. Commun. 47, 2014-2043 (2011)). We have also followed the same definition, and obtained the quantum efficiency of 1.5% for our catalyst.

Revision made:

Page 9: We added a sentence, “It is noted that 8 electrons are required for the production of one CH_4 molecule from CO_2 .”.

4. Line 196 indicates that water is oxidized in this reaction. However, no data is provided

showing the formation of O₂. This needs to be demonstrated if one wants to claim water oxidation.

Author response: Using the TCD detector in GC, we have detected the oxygen amount during the reaction. The amount of oxygen increased along the reaction progress for 3 h. However, the oxygen evolution was slower than what we expected in the ideal reaction. We think that photogenerated holes were partly consumed by the oxidation of the Cu₂O domains. The OH radicals, generated by the holes with water, may also react with the active surfaces. The actual role of the OH radicals in the possible reactions remains unclear, as some literatures described (Grimes et al. ACS Nano 2010, 4, 1259-1278 (2010)).

Revision made:

Page 14: We omitted the description of oxygen formation, because the discussion of the stability issue was completely changed (see below).

5. Lines 201-204, The selectivity of the process for methane is impressive, but it is not due to the indicated conduction band position.

Author response: For the issue of selectivity, we agree the reviewer's opinion, in which the product selectivity is limited by the energies of intermediates with kinetic factors, but is not determined by thermodynamic conduction band position. We have begun to study gas-phase reaction conditions using the present ZnO-Cu₂O catalysts, and as a preliminary result, the CH₄ generation was significantly diminished with the large production of CO when the small amount of water was used. This result indicates that water as a reaction medium behaves as a rich hydrogen source, leading to nearly quantitative production of CH₄ over other products. It is also reported that adsorbed CO are particularly stabilized on the Cu₂O(100) surface, which may allow more efficient coupling with adsorbed protons during the reaction. We have inserted the discussion of selectivity in the main manuscript.

Revision made:

Page 10: Figure 3e was changed into the proper diagram depicting bandgap alignment and reaction pathways.

Figure 3 | (e) Band alignment and proposed electron transfer mechanism of the ZnO-Cu₂O hybrid catalysts.

Pages 10-12: The section of “Mechanistic aspects of CO₂ conversion reactions” was completely re-written.

Pages 12-13: We inserted a paragraph as “For the issue of selectivity, this photocatalytic system provides sufficient energy based on the Z-scheme with photoexcited electrons at a high energy level to CO₂ reduction. It is known that the products are highly dependent upon relative energy levels of intermediates. In the present reaction system, the reaction medium, water, with a high proton concentration behaves a rich hydrogen source, and leads to the production of CH₄ more effectively. In addition, it is reported that the intermediates such as CO are particularly stabilized on the Cu₂O(100) surface, which may allow more efficient coupling with adsorbed protons during the reaction.”.

6. Line 237-8, the authors state, “Regarding direct CO₂ conversion, the ZnO-Cu₂O catalysts are definitely superior to TiO₂-Cu₂O for both reaction activity and selectivity.” But, in fact, what the data proves is that their catalyst is an improvement over P25-Cu₂O only.

Author response: As the reviewer recommended, we have used the term “TiO₂(P25)-Cu₂O” instead of “TiO₂-Cu₂O”.

Revision made:

Pages 13-14: All terms were changed into “TiO₂(P25)-Cu₂O”.

Page 18: We changed the term into “TiO₂(P25)-Cu₂O”.

7. Line 252-256, This is a false statement. Neither the reduction of CO₂ to methane or the oxidation of water to O₂ is reversible by any stretch of the imagination, under any set of conditions present in the experiment under consideration. In fact, these reactions are often used to illustrate the prototypically irreversible reaction.

8. Line 257-258, Why do the authors believe that O₂ is oxidizing the Cu₂O phase instead of this phase being oxidized by the photogenerated holes?

Author response: We deeply thank the reviewer for these valuable comments. In Fig. 5, we reported some saturation behaviors of the reactions by prolonged irradiation, which were explained by the reaction equilibrium between the reactants and products. However, as the reviewer claimed, we were totally incorrect, because neither CO₂ reduction into methane nor water oxidation are reversible reactions. We think that the decrease of the CH₄ production at 10-14 h is mainly due to the depletion of CO₂. To prove it, we have carried out more stability tests. In this experiment, the catalyst particles were re-dispersed in the fresh reaction medium with 0.2 M Na₂CO₃ after the first 4 h reaction, and the reaction was carried out. This process was repeated again to show the catalyst stability. As shown in the figure, the catalytic activity was regenerated after the use of the fresh reaction medium.

Based on these tests, we concluded that the catalyst stability maintains during the reaction for more than 12 h, when the fresh reaction medium is supplied. The present reaction proceeded inside the closed chamber, resulting in the CO₂ depletion along the prolonged reaction time. This would be a main cause for the significant decrease of the activity at the late stage of the reaction. The CO₂ depletion may change the pH of the reaction medium and slow down the CO₂ reduction reaction, which leads to the Cu₂O oxidation into CuO by photogenerated holes. We have added the data in the main text, and rewritten the section of “Stability of the ZnO-Cu₂O catalysts”.

Revision made:

Pages 14-15: The section of “Stability of the ZnO-Cu₂O catalysts” was completely re-written. Fig. 5b was changed as follows.

Figure 5 | Stability experiment of the ZnO-Cu₂O catalysts. (a) using the ZnO-Cu₂O catalysts as a function of the irradiation time up to 14 h. The reaction conditions were pH = 7.4 and λ > 200 nm. The CH₄ amount was converted based on the catalyst amount fixed to 19 mg. (b) The amount of CH₄ production under the identical reaction conditions except the change of the reaction medium at each 4 h reaction time.

It was a big challenge for us to address every issue that the reviewers raised, but we think we successfully reached the final goal. During this revision, we have learned many important things in photocatalytic CO₂ conversion, and we deeply respect the reviewers for their valuable comments and recommendations. We hope that we correctly represent our responses to the reviewers' concerns. Please kindly consider our revision that has been rationally prepared to provide a better manuscript suitable for the journal. Thank you for your cooperation on this matter.

Reviewer #1 (Remarks to the Author):

I have read the revised version of the manuscript “ZnO-Cu₂O Colloidal Nanocatalysts for Highly Selective Photocatalytic CO₂ Conversion into Methane by Water” by Bae et al together with its associated reply letter. In my opinion this version offers a substantial improvement over its predecessor, with most points addressed correctly.

My main concern remains the mechanism. The authors have accepted my criticism to the original version that the CO₂ reduction cannot happen on ZnO, but rather on Cu₂O. However, I am very skeptical about the Z-scheme proposed by the authors. In a Z-Scheme (reviewed e.g. by Maeda, ACS Catalysis 2013, 3, 1486-1503) the two absorbers are usually separated. A mediator redox pair is then employed to transfer the electrons from the oxidation component to the reduction component and to prevent the reverse process. In the system proposed by the authors, with the very good contact between the ZnO and Cu₂O parts, I do not see why the electron would not transfer from Cu₂O to ZnO (assuming the alignment as in Fig 3e). This would be expected to be much quicker than the CO₂ electrocatalytic reaction at the surface.

In addition the time-resolved PL measurements also suggest that the recombination in a single component is faster than in the composite which would also suggest that the proposed recombination (Fig.3e) between CB of ZnO and VB of ZnO is not that fast.

In effect, I find the proposed mechanism to be very unlikely. The authors would need to provide considerable more evidence that the other electron transfer processes are not happening.

That said, I find the results – in particular the efficiency – to be very interesting and worthy of publication at high level. I suggest that the authors try to perform the photocatalytic experiments using only visible light (to exclude absorption by ZnO) to verify whether absorption by ZnO is required or whether absorption by CuO is sufficient (adjusting the light intensity for comparable photon flux). Additionally, light intensity could also help to determine whether the reaction is a two- or single-photon process. The former would be expected in a Z-scheme.

The authors characterized the samples by XPS measurements prior to the irradiation. Could such measurement be performed on a sample after the irradiation? This could help establish whether a Cu(I)/Cu(II) junction is formed, perhaps facilitated by the ZnO phase.

I have two more comments on the manuscript:

1. The time-resolved PI decay curves should be presented on a scale of several nanoseconds if the lifetimes are in hundreds of ps (cf. Fig. 3d). It does not make sense to present the time until 1000ns in such case.

2. The arguments for the selectivity towards CH₄ over CO are rather weak in the paper. The additional experiments would probably be outside the scope, but I would suggest extending the discussion based on available literature on CO₂ reduction on copper(I) surfaces.

In summary, the manuscript is improved and the reported efficiency is notable, but the mechanism appears to be wrong. I suggest that the paper is not accepted at this time, but a further revision based on the experiments suggested above (and additional ones, if necessary to prove the point) could bring the paper to the level suitable for Nature Communications.

Reviewer #2 (Remarks to the Author):

In my previous report, I did not recommend publication in Nature Communications of this submission, since reports in the literature clearly indicate the instability of the material over the time, due to oxidation of Cu₂O and corrosion of ZnO. Now the authors have addressed this issue and, particularly corrected Figure 5, by performing a reuse and making other tests. I still have some concerns, but I have to recognize that the authors have presented now data in support of photocatalyst stability. In view of this as well as the answers to the other reports, now I recommend publication with minor changes. The authors still have to address the following comments:

- The reasons why the photocatalytic activity upon reuse increases have to be explained.
- The authors claim that the decrease in photocatalytic activity over 14 h is due to substrate depletion. Considering that CO₂ is in a very large excess, this reason seems unlikely. The authors have to provide CO₂ conversion at 14 h to convince the reader of CO₂ depletion.
- Figure showing the influence of ZnO/Cu₂O ratio on the photocatalytic activity that is provided in the author's answer has to be included in the manuscript, at least in the supplementary information and mentioned in the text.
- ¹³C-labelled CO₂ experiment needs to be presented better. Instead of stating that the m/z peak at 17 increases a lot, the percentage of ¹³CH₄ vs. ¹²CH₄ has to be quantified and indicated in the text. Also Figure S2 has to be expanded showing the mass spectra of the ¹³CO₂ labelled experiment.

- Visible light photoresponse has to be completed by adding the CH₄ production under these conditions.

Once these changes are made, publication would be recommended.

Reviewer #3 (Remarks to the Author):

The authors have carefully addressed all of the concerns of the referees. Additional experiments have been incorporated that address key open issues in the original manuscript, and conclusions have been adjusted to include the new information. Based on this analysis and my original conclusion that the manuscript reported material that was publishable, I now recommend publication of the paper. I do note that there are several small grammatical errors that should be corrected to improve the readability of the paper.

For Reviewer #1:

I have read the revised version of the manuscript “ZnO-Cu₂O Colloidal Nanocatalysts for Highly Selective Photocatalytic CO₂ Conversion into Methane by Water” by Bae et al together with its associated reply letter. In my opinion this version offers a substantial improvement over its processor, with most points addressed correctly.

Author response: We deeply appreciate the reviewer’s opinion that the previous revision addressed most of the issues properly.

1. My main concern remains the mechanism. The authors have accepted my criticism to the original version that the CO₂ reduction cannot happen on ZnO, but rather on Cu₂O. However, I am very skeptical about the Z-scheme proposed by the authors. In a Z-Scheme (reviewed e.g. by Maeda, ACS Catalysis 2013, 3, 1486-1503) the two absorbers are usually separated. A mediator redox pair is then employed to transfer the electrons from the oxidation component to the reduction component and to prevent the reverse process. In the system proposed by the authors, with the very good contact between the ZnO and Cu₂O parts, I do not see why the electron would not transfer from Cu₂O to ZnO (assuming the alignment as in Fig 3e). This would be expected to be much quicker than the CO₂ electrocatalytic reaction at the surface.

Author response: As we modified our previous manuscript, we totally agreed the reviewer’s opinion describing the fact that Cu₂O is an active surface for CO₂ reduction, and changed our explanation into Z-scheme-like mechanism. In fact, the Z-scheme mechanisms have been generally used for the systems with separated absorbers in the presence of mediators, however, the Z-schemes having intimately contacted semiconductors without mediators have also been reported. (Review: Jaroniec et al. Adv. Mater. 26, 4920-4935 (2014)). Domen et al. reported doped BiVO₄ and SrTiO₃ heterostructures for photocatalytic reactions, and proposed the Z-scheme mechanism (Nature Mater. 15, 611-615 (2016); J. Am. Chem. Soc. 139, 1675-1683 (2017)). Grimes et al. suggested the Z-scheme for the CO₂ reduction into CO in a TiO₂-CuO system in their review paper (ACS Nano 3, 1259-1278 (2010)). Recently, Li et al. compared between two opposite schemes of double-charge transfer and Z-scheme mechanisms on the Fe₂O₃/Cu₂O heterostructures, and concluded that the direct Z-scheme mechanism was more reasonable in their system (ACS Appl. Mater. Interfaces 7, 8631-8639 (2015)). It is noted that when a photocatalyst is immersed in water, charge transfer occurs at the semiconductor-solution interface due to the equilibration of electron density between two phases. The net result is the formation of an electrical field at the semiconductor surface, which leads to the hole transfer to the surface oxidizing water in n-type semiconductors (ZnO) when photogenerated electron-hole pairs forms in the space charge region. Similarly, photogenerated electrons move to the surface reducing CO₂ in p-type semiconductors (Cu₂O). (Nozik et al. Appl. Phys. Lett. 30, 567-569 (1977); Walter et al. Chem. Rev. 10, 6446-6473 (2010)).

For the facile electron transfer from ZnO to Cu₂O, we think that the excited electrons in the ZnO domains are trapped on the interface states, which may reduce the charge recombination rate to the valence band of ZnO and facilitate the tunneling to the valence band of Cu₂O (see below for responses 2 and 3). We have added the comments for the comparison between double charge transfer and Z-scheme mechanisms in the manuscript.

Revision made:

Page 12, Lines 8-15: Some sentences were modified as “Second, the formation of uniform

domain structures facilitates highly efficient electron and hole transfers to the reagents. When a photocatalyst is immersed in water, charge transfer occurs at the semiconductor-solution interface due to the equilibration of electron density between two phases. The net result is the formation of an electrical field at the semiconductor surface, which leads to the hole transfer to the surface oxidizing water in n-type semiconductors (ZnO), when photogenerated electron-hole pairs forms in the space charge region. Similarly, photogenerated electrons move to the surface reducing CO₂ in p-type semiconductors (Cu₂O).” with proper citations.

Page 13, Lines 3-13: A paragraph was added to explain the reaction mechanism as “The other mechanism, double charge transfer, which includes electron transfer from the conduction band of Cu₂O to ZnO domains and hole transfer from the valence band of ZnO to Cu₂O, has also been proposed in several photoreduction systems. However, in our catalysts, the CH₄ production of the pure ZnO aggregates was negligible, while the pure Cu₂O nanoparticles showed a significant activity (Figure 3c), indicating that the Cu₂O domains are main active sites for CO₂ reduction. In the aspect of band edge energies, CO₂ reduction needs sufficient energy over its electrochemical potential, and water oxidation also requires a large overpotential in general. Therefore, the Z-scheme mechanism in Figure 3e is more reasonable, where electron transfer occurs from the high-lying conduction band of the Cu₂O domains to CO₂ molecules. The low-lying valence band of the ZnO domains is also able to provide a sufficient overpotential for water oxidation reactions.” with proper citations.

2. In addition the time-resolved PL measurements also suggest that the recombination in a single component is faster than in the composite which would also suggest that the proposed recombination (Fig.3e) between CB of ZnO and VB of ZnO is not that fast.

Author response: Indeed, a similar time-resolved PL behavior was also observed in ZnO islands on CuO nanowires (Biswas et al. ACS Appl. Mater. Interfaces 7, 5685-5692 (2015)). In this literature, the authors explained the long photoexcited electron lifetime by the electron transfer from CuO to ZnO domains, which is the mechanism that we have originally proposed in our previous manuscript. However, as you commented, this double charge transfer mechanism has many problems unable to explain the negligible activity of ZnO and insufficient band edge energies to supply overpotentials of the redox reactions. Instead, we agree your and other reviewers’ suggestions of the Z-scheme mechanism, which is more reasonable for our catalytic system (also see below for response 3). For the long lifetime of photoexcited electrons, it is known that interface states are generated between the Cu₂O and ZnO domains, which trap charge carriers. In fact, the peak centered at 620 nm used for the PL measurement appears only in the spectrum of the ZnO-Cu₂O nanoparticles, but not in those of the pure ZnO and Cu₂O nanoparticles. This indicates that the 620 nm peak may root from the interfacial energy band transition, which was already observed in the Cu₂O/ZnO heterojunction nanostructure (Yu et al. Nanoscale 4, 7817-7824 (2012); Schmidt-Mende et al. Adv. Funct. Mater. 21, 573-582 (2011)). These trap levels may reduce the charge recombination rate from the conduction band to the valence band of ZnO, and facilitate the tunneling to the valence band of Cu₂O. To understand the photophysical mechanism in detail, further study is required.

Revision made:

Page 9, Lines 17-21: Some sentences were modified as, “On the other hand, the pure ZnO aggregates exhibit only weak signals in this region, and the Cu₂O nanoparticles have a distinct peak at 570 nm. Therefore, the peak centred at 620 nm roots from the interfacial

energy band transition, as observed in the Cu₂O/ZnO heterojunction nanostructure. The decay of transient absorption was measured at this wavelength.”.

Page 10, Lines 1-3: A sentence, “This may be attributed to the interface states trapping electrons, which reduce the charge recombination rate to from the conduction band to the valence band of ZnO and facilitate tunnelling to the valence band of Cu₂O.” was added with the citation of references 31 and 32.

3. In effect, I find the proposed mechanism to be very unlikely. The authors would need to provide considerable more evidence that the other electron transfer processes are not happening. That said, I find the results – in particular the efficiency – to be very interesting and worthy of publication at high level. I suggest that the authors try to perform the photocatalytic experiments using only visible light (to exclude absorption by ZnO) to verify whether absorption by ZnO is required or whether absorption by CuO is sufficient (adjusting the light intensity for comparable photon flux). Additionally, light intensity could also help to determine whether the reaction is a two- or single-photon process. The former would be expected in a Z-scheme. The authors characterized the samples by XPS measurements prior to the irradiation. Could such measurement be performed on a sample after the irradiation? This could help establish whether a Cu(I)/Cu(II) junction is formed, perhaps facilitated by the ZnO phase.

Author response: We sincerely thank to the reviewer for this valuable comment. The reviewer’s suggestion would be a direct evidence to prove the mechanism. When we irradiated visible light with a UV cut off filter ($\lambda > 420$ nm), CH₄ production was nearly negligible, and after the removal of the filter with a fixed light intensity (0.59 Wcm⁻²), CH₄ was generated with the activity similar to the previous experiment. There was no change of the surface states in the XPS spectrum after the visible light irradiation. This result indicates that the excitation of electrons in the ZnO domain is critical to activate the catalyst, and demonstrates that the Z-scheme is a reliable photochemical reaction mechanism. We have inserted this experimental result in the main text and supplementary information.

Supplementary Figure 6 | Photocatalytic reactions by the irradiation of visible light. Amount of CH₄ production under the irradiation of visible light with a UV cut-off filter ($\lambda >$

420 nm). After 6.5 h irradiation, the filter was removed. The light intensity before and after the removal of the filter was fixed at 0.59 Wcm^{-2} by adjusting the distance between the light source and the reactor.

Revision made:

Page 13, Lines 13-21: The sentences were added as “To prove the proper photophysical mechanism, the reaction was carried out by the irradiation of visible light under the present condition. The CH_4 production was almost negligible by the light irradiation with a UV cut-off filter ($\lambda > 420 \text{ nm}$), and the surface state of the catalyst was unchanged after the reaction. However, CH_4 was generated with the activity similar to that of the original experiment after the removal of the cut-off filter at a fixed light intensity of 0.59 Wcm^{-2} (Supplementary Figure 6). This result demonstrates that the excitation of electrons in the ZnO domain is critical to activate the catalyst, and consequently, Z-scheme is a more reliable reaction mechanism in our catalytic system.”.

Supplementary Figure 6 was inserted in Supplementary Information.

4. I have two more comments on the manuscript: The time-resolved PL decay curves should be presented on a scale of several nanoseconds if the lifetimes are in hundreds of ps (cf. Fig. 3d). It does not make sense to present the time until 1000ns in such case.

Author response: As the reviewer recommended, we have inserted expanded spectra in the region of 0 – 4 ns as an inset in Figure 3d.

Revision made:

Figure 3d was changed with the inclusion of spectra in the region of 0 – 4 ns.

5. The arguments for the selectivity towards CH_4 over CO are rather weak in the paper. The additional experiments would probably be outside the scope, but I would suggest extending the discussion based on available literature on CO_2 reduction on copper(I) surfaces.

Author response: For the issue of selectivity, there are a few proposed mechanisms, such as formaldehyde and carbene pathways. In our experiment, the selectivity of CH₄ is remarkably over 99%; therefore, it is hard to distinguish any of the mechanisms. The more important point is that the intermediates are strongly bound to the Cu₂O(100) surface, which induce the efficient reduction of CO into CH₄ with a rich hydrogen source of aqueous solutions at neutral pH. As the reviewer recommended, we extend the discussion of CO₂ reduction based on commonly acceptable mechanisms with adding the comments of the uniqueness of the Cu₂O(100) surface.

Revision made:

Page 14, Lines 1-12: The description of the reaction mechanism was inserted as “Gattrell and many other researchers suggested that the radical anion of CO₂ is adsorbed on the metal surface and forms a carboxylic radical, which converts to CO by the interaction with surface hydrogen radical. According to the calculations, the rate determining step of the process is the hydrogenation of CO into the formyl radical, which majorly decides the product distribution. Cu has a strong binding strength for adsorbed intermediates and facilitate the hydrogenation. More specifically, it is reported that the intermediates are particularly stabilized on the Cu₂O(100) surface, which prevents the desorption of CO and allow efficient coupling with protons during the reaction. In the present reaction conditions, the reaction medium, water, with a high proton concentration at neutral pH behaves a rich hydrogen source and directly supplies protons. The resulting intermediates, such as formyl radicals or carbenes, are further hydrogenized to produce CH₄ eventually. To understand the reaction mechanism in detail, however, further study is demanded.” with proper citations of references 41-44.

For Reviewer #2:

In my previous report, I did not recommend publication in Nature Communications of this submission, since reports in the literature clearly indicate the instability of the material over the time, due to oxidation of Cu₂O and corrosion of ZnO. Now the authors have addressed this issue and, particularly corrected Figure 5, by performing a reuse and making other tests. I still have some concerns, but I have to recognize that the authors have presented now data in support of photocatalyst stability. In view of this as well as the answers to the other reports, now I recommend publication with minor changes. The authors still have to address the following comments:

Author response: We deeply appreciate the reviewer’s opinion that the previous revision was effective to prove the photocatalyst stability.

1. The reasons why the photocatalytic activity upon reuse increases have to be explained.

Author response: We think that the photocatalytic activity partly depends on the reaction environment. In the case of recycle at each 4 h reaction time as shown in Fig. 5b, the activity did not change a lot between the cycles. In the continuous 14 h reaction periods, we expect that some surfactants were presumably detached from the catalysts, generating naked active Cu₂O surface to generate more CH₄.

2. The authors claim that the decrease in photocatalytic activity over 14 h is due to substrate depletion. Considering that CO₂ is in a very large excess, this reason seems unlikely. The authors have to provide CO₂ conversion at 14 h to convince the reader of CO₂ depletion.

Author response: To address the reviewer's concern, we have calculated CO₂ conversion based on the amount of dissolved CO₂ in water using the following equations (Ridgwell et al. Nature Education Knowledge 3, 21 (2012); Dickson et al. Marine Chem. 106, 287-300 (2007)).

$$\text{Conversion (\%)} = \frac{\text{The amount of generated CH}_4 \text{ (mol)}}{\text{The amount of CO}_2 \text{ in water (mol)}} \times 100$$

$$[\text{CO}_2]_{eq} = \frac{[\text{H}^+]_{eq}^2}{[\text{H}^+]_{eq}^2 + K_1[\text{H}^+]_{eq} + K_1K_2} \times \text{DIC}$$

$$[\text{HCO}_3^-]_{eq} = \frac{K_1[\text{H}^+]_{eq}}{[\text{H}^+]_{eq}^2 + K_1[\text{H}^+]_{eq} + K_1K_2} \times \text{DIC}$$

$$[\text{CO}_3^{2-}]_{eq} = \frac{K_1K_2}{[\text{H}^+]_{eq}^2 + K_1[\text{H}^+]_{eq} + K_1K_2} \times \text{DIC}$$

$$\text{DIC (Total dissolved inorganic carbon)} = [\text{CO}_{2(aq)}] + [\text{HCO}_3^-] + [\text{CO}_3^{2-}]$$

As a result, the CO₂ conversion is estimated as 47% after the 14 h irradiation of light. The CO₂ depletion may be compensated from CO₂ gas inside the chamber, but in the real experiment, the CO₂ concentration in water may decrease further due to the temperature rising by prolonged irradiation. Severe decrease of the reactivity due to the transfer of dissolved CO₂ to the gas phase was also reported in the literature (Choi et al. Energy Environ. Sci. 5, 6066-6070 (2012)).

Revision made:

Page 16, Lines 1-3: A sentence was modified as “The reaction is carried out in a closed chamber; therefore, the CO₂ depletion in the reaction medium may be the main reason for this activity decrease (See Supplementary Information).”.

Supplementary Information: The section of “Calculation of CO₂ Conversion” was added with the proper citations.

3. Figure showing the influence of ZnO/Cu₂O ratio on the photocatalytic activity that is provided in the author's answer has to be included in the manuscript, at least in the

supplementary information and mentioned in the text.

Author response: As the reviewer recommended, we have inserted a paragraph describing the ZnO-Cu₂O catalysts having different Zn/Cu ratios, and added the CH₄ production graph in Supplementary Information.

Supplementary Figure 5 | CH₄ production using the catalysts with different Zn/Cu ratios. Amount of CH₄ production using the ZnO-Cu₂O catalysts synthesized from the Zn and Cu precursor ratios of 2:1 (red), 1:1 (blue, optimized structure), and 2:3 (green), and using pure ZnO aggregates (black).

Revision made:

Page 10, Lines 10-14: A paragraph was inserted as “The CH₄ production rates were also measured using ZnO-Cu₂O catalysts synthesized from the various ratio of the Zn/Cu precursors, but the activities were inferior to that of the optimized catalyst (Supplementary Fig. 5). It is because either the Cu₂O domains were not fully grown on the ZnO surface, or the resulting catalyst was not uniform in its morphology. This indicates that the catalyst structure is an essential factor to maximize the catalytic performances.”

Supplementary Figure 5 was added in Supplementary Information.

4. 13-labelled CO₂ experiment needs to be presented better. Instead of stating that the m/z peak at 17 increases a lot, the percentage of ¹³CH₄ vs. ¹²CH₄ has to be quantified and indicated in the text. Also Figure S2 has to be expanded showing the mass spectra of the ¹³CO₂ labelled experiment.

Author response: In the isotope experiment, mostly ¹³CH₄ but also a small amount of ¹³CO₂ were produced. It was very difficult to isolate the only CH₄ signals in our GC-MS instrument, because the fragmentation peaks from small molecules such as CH₄ and H₂O usually appeared together in the early region of the spectrum. Instead, the peak at m/e = 17 significantly increased when the ¹³CO was used as shown in Supplementary Fig. 2. In addition, although CO was produced in a small amount in the reaction, the ¹³CO peak (m/z = 29) uniquely appeared when ¹³CO₂ was used for the reaction. Based on the control experiment with N₂ and the large amount generation of CH₄ by prolonged irradiation, we ensure that CH₄ was actually generated from CO₂ reduction.

5. Visible light photoresponse has to be completed by adding the CH₄ production under these conditions.

Author response: We thank to the reviewer for this valuable comment. As the reviewer recommended, we have carried out the visible light experiment. When we irradiated visible light with a UV cut off filter ($\lambda > 420$ nm), CH₄ production was nearly negligible, and after the removal of the filter with a fixed light intensity (0.59 Wcm^{-2}), CH₄ was generated with the activity similar to the previous experiment. There was no change of the surface states in the XPS spectrum after the irradiation. This result indicates that the excitation of electrons in the ZnO domain is critical to activate the catalyst, and demonstrates that the Z-scheme is a reliable photochemical reaction mechanism. We have inserted this experimental result in the main text and supplementary information.

Revision made:

Page 13, Lines 13-21: The sentences were added as “To prove the proper photophysical mechanism, the reaction was carried out by the irradiation of visible light under the present condition. The CH₄ production was almost negligible by the light irradiation with a UV cut-off filter ($\lambda > 420$ nm), and the surface state of the catalyst was unchanged after the reaction. However, CH₄ was generated with the activity similar to that of the original experiment after the removal of the cut-off filter at a fixed light intensity of 0.59 Wcm^{-2} (Supplementary Figure 6). This result demonstrates that the excitation of electrons in the ZnO domain is critical to activate the catalyst, and consequently, Z-scheme is a more reliable reaction mechanism in our catalytic system.”.

Supplementary Figure 6 was inserted in Supplementary Information.

Supplementary Figure 6 | Photocatalytic reactions by the irradiation of visible light.

Amount of CH₄ production under the irradiation of visible light with a UV cut-off filter ($\lambda > 420$ nm). After 6.5 h irradiation, the filter was removed. The light intensity before and after the removal of the filter was fixed at 0.59 Wcm^{-2} by adjusting the distance between the light source and the reactor.

For Reviewer #3:

The authors have carefully addressed all of the concerns of the referees. Additional experiments have been incorporated that address key open issues in the original manuscript, and conclusions have been adjusted to include the new information. Based on this analysis and my original conclusion that the manuscript reported material that was publishable, I now recommend publication of the paper. I do note that there are several small grammatical errors that should be corrected to improve the readability of the paper.

Author response: We deeply appreciate the reviewer's conclusion that now the manuscript is acceptable in this journal. We have corrected our manuscript by the help of a professional English-native editor.

Reviewer #1 (Remarks to the Author):

I have re-read the revised manuscript and the replies provided by the authors. My main concern related the mechanism of the reaction, in particular the proposed Z-scheme. I agree with the authors that the typical Type II transfer (called double charge transfer by the authors) cannot explain the results, as I have argued in my original report. In the Z-scheme, the alignment would be appropriate with CO₂ reduction on the Cu₂O phase and oxidation on the ZnO phase, but the major difficulty is proving why electrons transfer from CB of ZnO to VB of Cu₂O, rather than from CB of Cu₂O to CB of ZnO. The latter is normally expected in Type II band alignment. In the revised manuscript the authors have argued this on the basis intimate contact between the Cu₂O and ZnO phases and charge distribution (band bending) at the solid-liquid interface. They also provide some additional experimental evidence (transient absorption, photocatalytic measurements at visible light illumination only, etc.). I am not entirely convinced by the explanation, but I accept that it is in agreement with the observations. I suspect that some specific trap states at the ZnO-Cu₂O interface facilitate the electron transfer in the way that fits the Z-scheme, as implied by the PL and TA results. In the current stage, I recommend that the authors use a cautious language when describing the mechanism (suggest, indicate rather than prove). A further brief discussion on this would be advisable, in my opinion, by I understand that it can only be speculative in nature.

I commend the inclusion of the ¹³C experiments, but a phrase “increased a lot” is not suitable without quantification. The results in SI allow for such quantification and the knowledge of the CH₄ breaking pattern in the MS (easily approximated by measuring it for pure CH₄ in the MS) could even allow for estimating the percentage of final CH₄ originating from CO₂.

Minor comments:

1. Equation 1: 8 x number of CH₄ molecules, not number of CH₄
2. 'It is reported that the Zn precursors...' – should read 'It is reported here that...'
3. 'It is known the photoexcited electrons are trapped in the conduction band on a time scale of hundred fs' – Do the authors refer to thermalization of the hot carriers to the bottom edge of the conduction band or to trapping of the electrons below the conduction band in interband states (in shallow traps)?

4. For TiO₂-Cu₂O the actual rates should be provided (currently only 'total gas production was less than half')

5. There is no comment in the current manuscript on the oxidation reaction. I understand that this is not the objective of the paper, but a comment would be expected. There is no mention in the manuscript of hole scavengers, so either water is oxidized or – presumably – the PVP ligands act as a hole scavenger.

In summary, I find the manuscript to be improved over its previous iteration and I consider it suitable for publication in Nature Communications. I suggest however that the issues listed above are taken in to consideration in the preparation of the final version.

Reviewer #2 (Remarks to the Author):

The authors have gone through a lengthy review process and have addressed properly most of my previous comments. However, the requested isotopic experiment is not convincing in the way that is presented. The fact that the authors are not able to quantify the percentage of product having ¹³C with respect to ¹²C is clearly detrimental to provide a firm evidence of the origin of methane. Also the authors confuse the reader when addressing the conversion of CO₂, since they only consider for the calculation of conversion the dissolved CO₂ and not the total (dissolved and gaseous CO₂) and also consider that the equilibrium between dissolved and gas phase CO₂ is slow. Nevertheless, at this point, I think that the paper should be accepted.

We are sincerely thankful to the editor and reviewers for their generous help of making a better manuscript. We have spent two more months to try to satisfy the reviewers' additional recommendations. Our point-by-point response to each of the reviewers' revision requests are as follows. The corrections are marked in red in the main manuscript.

For Reviewer #1:

1. I have re-read the revised manuscript and the replies provided by the authors. My main concern related the mechanism of the reaction, in particular the proposed Z-scheme. I agree with the authors that the typical Type II transfer (called double charge transfer by the authors) cannot explain the results, as I have argued in my original report. In the Z-scheme, the alignment would be appropriate with CO₂ reduction on the Cu₂O phase and oxidation on the ZnO phase, but the major difficulty is proving why electrons transfer from CB of ZnO to VB of Cu₂O, rather than from CB of Cu₂O to CB of ZnO. The latter is normally expected in Type II band alignment. In the revised manuscript the authors have argued this on the basis intimate contact between the Cu₂O and ZnO phases and charge distribution (band bending) at the solid-liquid interface. They also provide some additional experimental evidence (transient absorption, photocatalytic measurements at visible light illumination only, etc.). I am not entirely convinced by the explanation, but I accept that it is in agreement with the observations. I suspect that some specific trap states at the ZnO-Cu₂O interface facilitate the electron transfer in the way that fits the Z-scheme, as implied by the PL and TA results. In the current stage, I recommend that the authors use a cautious language when describing the mechanism (suggest, indicate rather than prove). A further brief discussion on this would be advisable, in my opinion, by I understand that it can only be speculative in nature.

Author response: As we discussed in our previous response, the TCSPC results were rather controversial, but the other experimental data matched the Z-scheme-like mechanism in our catalytic system. We suggested that the interface trap levels may facilitate carrier tunneling to the interface for recombination between the holes in the VB of Cu₂O and the electrons in the CB of ZnO, based on the report on ZnO-Cu₂O photovoltaic cells (ref 32, Schmidt-Mende et al., *Adv. Funct. Mater.* 2011, 21, 573-582). In this reference, the authors observed the reduction of the built-in potential in the ZnO-Cu₂O diode. They suggested that the holes from the Cu₂O could easily reach the interface state and recombine with the electrons from the ZnO (in Figure 10b in ref 32). However, this is only a speculation; and thus, we rearranged the TCSPC data in the main text into Supporting Information, and carefully modified our discussion on this issue. Instead, the photoresponse data was moved to Figure 3d, because they indicate the importance of visible light absorption to generate photoelectrons.

Revision made:

Page 9, Lines 14-21: The description of the TCSPC measurement was shortened as “The lifetime of photogenerated electrons was directly measured by means of time-correlated single photon counting (TCSPC). The decay of transient absorption was measured at 620 nm, which roots from the interfacial energy band transition. The photoexcited electron lifetime of ZnO-Cu₂O nanoparticles ($\tau_{1/2} = 837.1$ ps) is large, compared to those of ZnO ($\tau_{1/2} = 491.4$ ps) and Cu₂O ($\tau_{1/2} = 206.5$ ps) nanoparticles (Supplementary Fig. 3). This may be attributed to the interface states trapping electrons, which reduce the charge recombination rate from the conduction band to the valence band of ZnO and facilitate tunnelling to the valence band of

Cu₂O.”

Page 10, Lines 13-17: The sentences were modified as “Well-defined domain structures are expected to induce a proper bandgap alignment as depicted in Fig. 3e. In the Z-scheme mechanism, the effective electron transfer from the conduction band of ZnO to the valence band of Cu₂O domains leads to long-lived charge separation states with the excited electrons at the conduction band of the Cu₂O domain and the holes at the valence band of the ZnO domain.”

Page 11, Figure 3d: The figure was changed by the photoresponse data.

Supplementary Fig. 3b: The figure was changed by the TCSPC spectra.

Page 13, Lines 6 and 11: The words were changed into “suggest” and “indicates”.

2. I commend the inclusion of the ¹³C experiments, but a phrase “increased a lot” is not suitable without quantification. The results in SI allow for such quantification and the knowledge of the CH₄ breaking pattern in the MS (easily approximated by measuring it for pure CH₄ in the MS) could even allow for estimating the percentage of final CH₄ originating from CO₂.

Author response: As the reviewer recommended, we calculated the percentage of CH₄ derived from CO₂ to be 88% by the integration of the signals in the chromatogram in Supplementary Figure 2.

Revision made:

Page 8, Line 4 – Page 9, Line 2: The sentence was modified as “Based on a signal at m/e = 17, assignable to the ¹³CH₄ peak in the gas chromatography-mass spectrometry (GC-MS) chromatogram when ¹³CO₂ and Na₂¹³CO₃ were used, the percentage of CH₄ directly generated from CO₂ was estimated to be 88% during the reaction.”.

Minor comments:

1. Equation 1: 8 x number of CH₄ molecules, not number of CH₄
2. 'It is reported that the Zn precursors...' – should read 'It is reported here that...'

Revision made:

Page 9, Line 5: Equation 1 was corrected with “number of CH₄ molecules”.

Page 6, Line 17: The sentence was changed into “It is known that...”.

3. 'It is known the photoexcited electrons are trapped in the conduction band on a time scale of hundred fs' – Do the authors refer to thermalization of the hot carriers to the bottom edge of the conduction band or to trapping of the electrons below the conduction band in interband states (in shallow traps)?

Author response: The meaning of the sentence is the former, thermalization of the hot

carriers to the conduction band edge within a time scale of hundred fs. However, this sentence was removed not to emphasize the discussion of the TCSPC data in the main text as in the response of the reviewer comment #1.

4. For TiO₂-Cu₂O the actual rates should be provided (currently only 'total gas production was less than half')

Author response: We thank the reviewer for this valuable comment. When the catalyst (9.5 mg) was used, the amounts of the gas products were 3.8, 0.28, and 17 μmol for CH₄, CO, and H₂, respectively, by the 3 h irradiation. These values were converted into the activities of 130, 10, and 580 $\mu\text{mol/g}_{\text{cat}}\text{h}$ for CH₄, CO, and H₂, respectively.

Revision made:

Page 14, Lines 16-21: The sentences were modified as “Under the present reaction conditions of the CO₂ reduction by the irradiation for 3 h, the amounts of the gas products using the TiO₂(P25)-Cu₂O catalysts (9.5 mg) were 3.8, 0.28, and 17 μmol for CH₄, CO, and H₂, respectively. The activity for each product was estimated as 130, 10, and 580 $\mu\text{mol/g}_{\text{cat}}\text{h}$ for CH₄, CO, and H₂, respectively, of which the total gas production was inferior to that using the ZnO-Cu₂O catalyst (Fig. 4c).”

5. There is no comment in the current manuscript on the oxidation reaction. I understand that this is not the objective of the paper, but a comment would be expected. There is no mention in the manuscript of hole scavengers, so either water is oxidized or – presumably – the PVP ligands act as a hole scavenger.

Author response: We have detected the oxygen amount during the reaction using the TCD detector in GC. However, the oxygen evolution was slower than what we expected from the reaction stoichiometry. As the reviewer suggested, the organic residues such as surfactants can also behave as a hole scavenger consuming the photogenerated holes during the early stage of the photocatalytic reaction.

Revision made:

Page 14, Lines 5-8: The discussion was added as “The counter reaction, oxidation, should occur by the photogenerated holes at the same time. Mostly the holes were transferred to water molecules and led to oxygen evolution, which were detectable by GC, but the PVP adsorbed on the catalyst surface might also behave as a hole scavenger during the early stage of the photocatalytic reaction.”

In summary, I find the manuscript to be improved over its previous iteration and I consider it suitable for publication in Nature Communications. I suggest however that the issues listed above are taken in to consideration in the preparation of the final version.

Author response: We deeply appreciate the reviewer’s effort giving very valuable comments and recommendations for our manuscript.

For Reviewer #2:

1. The authors have gone through a lengthy review process and have addressed properly most of my previous comments. However, the requested isotopic experiment is not convincing in the way that is presented. The fact that the authors are not able to quantify the percentage of product having ^{13}C with respect to ^{12}C is clearly detrimental to provide a firm evidence of the origin of methane.

Author response: As the reviewer recommended, we have tried different ways of the isotope experiment for additional two months. However, there have been serious difficulties to get the clear data, for example, the high concentration of water and the detection limit of the GC-MS equipment. We definitely observed a significant increment of the ^{13}C mass pattern, but the problem was that the pattern was very difficult to separate from the water fragmentation signals and the baseline. Instead, we observed a large increase of the $m/e = 17$ ($^{13}\text{CH}_4$) signal as shown in Supplementary Figure 2, and by comparison with the peak from natural CH_4 , we estimated the percentage of CH_4 derived from CO_2 to be 88%. We have also observed a single signal of ^{13}CO when using $^{13}\text{CO}_2$ and $\text{Na}_2^{13}\text{CO}_3$. As described in the main text, the N_2 bubbling experiment could not generate any gaseous product, and the regeneration experiment by the repeated supply of Na_2CO_3 continuously produced CH_4 in a nearly constant rate. We believe that all of these results are indicative of the production of CH_4 from CO_2 .

Revision made:

Page 8, Line 4 – Page 9, Line 2: The sentence was modified as “Based on a signal at $m/e = 17$, assignable to the $^{13}\text{CH}_4$ peak in the gas chromatography-mass spectrometry (GC-MS) chromatogram when $^{13}\text{CO}_2$ and $\text{Na}_2^{13}\text{CO}_3$ were used, the percentage of CH_4 directly generated from CO_2 was estimated to be 88% during the reaction.”.

2. Also the authors confuse the reader when addressing the conversion of CO_2 , since they only consider for the calculation of conversion the dissolved CO_2 and not the total (dissolved and gaseous CO_2) and also consider that the equilibrium between dissolved and gas phase CO_2 is slow. Nevertheless, at this point, I think that the paper should be accepted.

Author response: We sincerely thank the reviewer for the comment. As the reviewer recommended, we calculated the total amount of CO_2 including dissolved and gaseous CO_2 and proposed that all CO_2 contributed as a reactant. As a result, the conversion of CO_2 after the 14 h irradiation of light is estimated to be 41%, which is not significantly different from the previous calculation result.

Revision made:

Page 3 in Supporting Information: The sentences were modified as “When the total amount of CO_2 including dissolved and gaseous forms participated in the reaction, the CO_2 conversion is estimated as 41% after the 14 h irradiation of light.”.

We hope that we correctly represent our responses to the reviewers' concerns. Thank you for your cooperation on this matter.

Reviewer #1 (Remarks to the Author):

I am satisfied that the authors have convincingly addressed the remaining questions from the reviewers. I have only very minor corrections to suggest at this stage:

1. in response to Rev1, Q1 the phrase "roots from the interfacial energy band transition" should be modified. I suggest a change to e.g. "corresponds to transition between the energy bands at the interface" or clarifying the sentence in a different way.

2. in response to Rev1, Q1, the phrase "a proper bandgap alignment" should be replaced with "an appropriate bandgap alignment"

Other than these two, I consider that the manuscript has improved considerably over the latest (and earlier) revisions and recommend it to be accepted for publication by Nature Communications.

Reviewer #2 (Remarks to the Author):

The authors have now quantitatively provided the proportion of ^{13}C labelled CO_2 indicating that a large percentage of the CH_4 formed comes definitely from CO_2 . They have also modified the value of conversion by considering the total amount of CO_2 present (dissolved and in the gas phase). After these changes, publication in Nature Commun is recommended.

Dear Editor,

Re: Manuscript ID: NCOMMS-16-23004C

TITLE: Colloidal zinc oxide-copper(I) oxide nanocatalysts for selective aqueous photocatalytic carbon dioxide conversion into methane

We are sincerely thankful to the editor and reviewers on our manuscript. Our point-by-point response to each of the reviewers' revision requests are as follows.

For Reviewer #1:

I am satisfied that the authors have convincingly addressed the remaining questions from the reviewers. I have only very minor corrections to suggest at this stage:

1. in response to Rev1, Q1 the phrase "roots from the interfacial energy band transition" should be modified. I suggest a change to e.g. "corresponds to transition between the energy bands at the interface" or clarifying the sentence in a different way.
2. in response to Rev1, Q1, the phrase "a proper bandgap alignment" should be replaced with "an appropriate bandgap alignment"

Other than these two, I consider that the manuscript has improved considerably over the latest (and earlier) revisions and recommend it to be accepted for publication by Nature Communications.

Revision made:

We appreciate very much for the reviewer's final evaluation. We changed 1 and 2 as the reviewer recommended.

Page 9, Lines 17-18: The sentence was modified as "corresponds to transition between the energy bands at the interface".

Page 10, Lines 14-15: The sentence was modified as "an appropriate bandgap alignment".

For Reviewer #2:

The authors have now quantitatively provided the proportion of ¹³C labelled CO₂ indicating that a large percentage of the CH₄ formed comes definitely from CO₂. They have also modified the value of conversion by considering the total amount of CO₂ present (dissolved and in the gas phase). After these changes, publication in Nature Commun is recommended.

Author response: We sincerely thank the reviewer for the final evaluation on our manuscript.

Thank you for your cooperation on this matter.